# Cost-effectiveness of a patient-reported outcome-based remote monitoring and alert intervention for early detection of critical recovery after joint replacement: A randomised controlled trial

Lukas Schöner [1‡*], David Kuklinski [2‡*], Laura Wittich [1], Viktoria Steinbeck [1], Benedikt Langenberger [1], Thorben Breitkreuz [3], Felix Compes [4], Mathias Kretzler [5], Ursula Marschall [6], Wolfgang Klauser [7], Mustafa Citak [8], Georg Matziolis [9], Daniel Schrednitzki [10], Kim Grasböck [11], Justus Vogel [2], Christoph Pross [1], Reinhard Busse [1‡], Alexander Geissler [2‡]

1 Department of Health Care Management, Technische Universität Berlin, Berlin, Germany, 2 Chair of Health Economics, Policy and Management, School of Medicine, University of St. Gallen, St. Gallen Switzerland, 3 aQua-Institut, Göttingen, Germany, 4 Heartbeat Medical Solutions GmbH, Cologne, Germany, 5 BKK Dachverband e.V., Berlin, Germany, 6 BARMER Institut für Gesundheitsforschung, Wuppertal, Germany, 7 VAMED Ostseeklinik Damp, Damp, Germany, 8 Helios ENDO-Klinik, Hamburg, Germany, 9 University Hospital Jena, Campus Eisenberg, Germany, 10 Sana Kliniken Sommerfeld, Kremmen, Germany, 11 RoMed Klinik, Prien am Chiemsee, Germany

‡ LS and DK share first authorship on this work. RB and AG are joint senior authors on this work.
* Lukas.schoener@tu-berlin.de (LS); david.kuklinski@unisg.ch (DK)

## Abstract

### Background

While the effectiveness of patient-reported outcome measures (PROMs) as an intervention to impact patient pathways has been established for cancer care, it is unknown for other indications. We assessed the cost-effectiveness of a PROM-based monitoring and alert intervention for early detection of critical recovery paths following hip and knee replacement.

### Methods and findings

The cost-effectiveness analysis (CEA) is based on a multicentre randomised controlled trial encompassing 3,697 patients with hip replacement and 3,110 patients with knee replacement enrolled from 2019 to 2020 in 9 German hospitals. The analysis was conducted with a subset of 546 hip and 492 knee replacement cases with longitudinal cost data from 24 statutory health insurances. Patients were randomised 1:1 to a PROM-based remote monitoring and alert intervention or to a standard care group. All patients were assessed at 12-months post-surgery via digitally collected PROMs. Patients within the intervention group were additionally assessed at 1-, 3-, and 6-months post-surgery to be contacted in case of critical recovery paths. For the effect evaluation, a PROM-based composite measure (PRO-CM) was developed, combining changes across various PROMs in a single index ranging from 0 to 100. The PRO-CM included 6 PROMs focused on quality of life and various aspects of

**Data Availability Statement:** Since the data contains sensitive patient information it is legally not allowed to make the data publicly accessible to others due to the German data protection law and the data protection agreements within the trial (see S1 Appendix). In order to enable verifiability of the study results after completion of the project (09/2023), the data will be stored at the research institutions - TU Berlin and aQua Institute - for a period of 10 years after project completion. Data access can only be granted in exceptional cases. Requests for data access should be addressed to the Data Protection Officer at TU Berlin: Anette Hiller (info@datenschutz.tu-berlin.de).

**Funding:** The PROMoting Quality project was funded by the Innovation Fund of the of Joint Federal Committee Germany (https://innovationsfonds.g-ba.de/) under grant number 01NVF18016. The PROMoting Quality consortium lead was at The Department of Health Care Management at the Technische Universität Berlin, with RB as head of the department. Project funding was paid to the PROMoting Quality consortium institutions and covered the salaried employment positions of LS, VS, LW, BL and CP at the Technische Universität Berlin. The funders of the study had no role in study design, data collection, data analysis, data interpretation, or writing of the report.

**Competing interests:** The project was funded by the Innovation Fund of the of Joint Federal Committee Germany (01NVF18016). Project funding was paid to the consortium institutions and covered the employment positions of LS, VS, LW, BL and CP. CP reports a salaried position with MedTech company Stryker that is separate and independent from his university research employment and position. FC reports an employment relation with Heartbeat Medical Solutions GmBH. GM reports receiving royalties or contracts, grants, consulting fees, payments for lectures and leadership in other boards, all outside the submitted work. RB reports being member of the government commission on modern and needs-based hospital care, outside the submitted work. DS reports receiving payments for lectures from Zimmer Biomet outside the submitted work. All other authors declare no competing interests.

**Abbreviations:** ASA, American Society of Anesthesiologists; BMI, body mass index; CEA, cost-effectiveness analysis; CI, confidence interval; GP, general practitioner; HRQoL, health-related quality of life; ICER, incremental cost-effectiveness ratio; ICHOM, International Consortium for Health Outcomes Measurement; OECD, Organisation for Economic Co-operation and Development; PRO-

physical and mental health. The primary outcome was the incremental cost-effectiveness ratio (ICER). The intervention group showed incremental outcomes of 2.54 units PRO-CM (95% confidence interval (CI) [0.93, 4.14]; $p = 0.002$) for patients with hip and 0.87 (95% CI [−0.94, 2.67]; $p = 0.347$) for patients with knee replacement. Within the 12-months post-surgery period the intervention group had less costs of 376.43€ (95% CI [−639.74, −113.12]; $p = 0.005$) in patients with hip, and 375.50€ (95% CI [−767.40, 16.39]; $p = 0.060$) in patients with knee replacement, revealing a dominant ICER for both procedures. However, it remains unclear which step of the multistage intervention contributes most to the positive effect.

## Conclusions

The intervention significantly improved patient outcomes at lower costs in patients with hip replacements when compared with standard care. Further it showed a nonsignificant cost reduction in knee replacement patients. This reinforces the notion that PROMs can be utilised as a cost-effective instrument for remote monitoring in standard care settings.

## Trial registration

**Registration**: German Register for Clinical Studies (DRKS) under DRKS00019916.

---

## Author summary

### Why was this study done?

- Hip and knee replacements are high volume treatments in most Western countries and represent a substantial driver for national healthcare expenditures.

- Evidence on the cost-effectiveness of using patient-reported outcome measures (PROMs) for monitoring in orthopaedics is very limited.

- The increased use of PROMs after orthopaedic surgery, the lack of standardised recovery follow-ups post-surgery combined with positive evidence from other treatment areas motivated the design of the "PROMoting Quality" trial.

### What did the researchers do and find?

- We performed a cost-effectiveness analysis (CEA) based on a multicentre randomised controlled trial encompassing 546 patients with hip and 492 patients with knee replacement enrolled from 2019 to 2020 in 9 German hospitals.

- Patients were randomised to a PROM-based remote monitoring and alert intervention for early detection of critical recovery paths, or to a standard care group without monitoring.

- The intervention improved patient outcomes in hip replacements when compared with standard care and showed a cost reduction in both patients with hip and knee replacement.

CM, PRO composite measure; PROM, patient-reported outcome measure; RCT, randomised controlled trial; SD, standard deviation.

### What do these findings mean?

- Undetected critical recovery after surgery result in both undesired health outcomes and increased health care expenditures.

- In times of resource constraints and staff shortages, evidence from the "PROMoting Quality" trial can inform medical professionals and policy makers that the use of a PROM monitoring and alert system post-surgery can help to improve health outcomes for patients with hip replacement and reduce healthcare expenditures for both procedures, and thus be implemented on system level.

- It can give guidance to the discussions about advancing the digitalisation of the health care sector and further initiates research of using PROMs as monitoring and alert system for other conditions such as chronic diseases. However, it remains unclear exactly which step of the multistage intervention causes the positive effect.

## Introduction

Hip and knee replacement surgery is considered an effective treatment option for patients suffering from hip or knee osteoarthritis [1]. In recent years, the complementary use of patient-reported outcome measures (PROMs) for evaluating treatment outcome has become more common [2] and influential [3,4]. While traditionally success of hip and knee replacement surgery was measured by outcome indicators such as complications or revision rates, PROMs refocus outcome assessment on the patient perspective by measuring, e.g., health-related quality of life (HRQoL), pain, and functional status [5].

These instruments have confirmed the effectiveness of hip and knee replacement surgery [6,7], but also showed that a significant number of patients rate their surgery as not beneficial. Kahlenberg and colleagues [8] report that around 15% to 20% of patients are dissatisfied with their results, with non-optimal postoperative functional outcome and lasting pain in the joint being the main determinants.

In Western countries, there is a high rate of hip and knee replacements. For example, in Germany, Pabinger and colleagues [9,10] reported intervention rates of 280 (hip replacement) and 230 (knee replacement) per 100,000 population, compared to the Organisation for Economic Co-operation and Development (OECD) average of 160 hip and 150 knee replacements per 100,000 population [11]. Furthermore, various projections report further increases of primary and revision procedures in Western countries [3,4], driven by an ageing population, an increasing number of younger patients and general health system advancements [12,13]. This places a substantial financial strain on national healthcare budgets.

One potential lever to improve mid-to-long-term outcomes, and in turn decrease follow-up treatments, is to monitor patients via PROMs post-surgery to detect critical recovery pathways and, if necessary, adapt treatment protocols [14]. This concept is based on evidence gathered from studies in the field of oncology that showed that regular monitoring of cancer patients via PROMs improved HRQoL, decreased the number of hospital admissions and emergency room visits, and increased the length of overall survival [15,16]. In further oncological studies, it was shown that the intervention was cost-effective [17,18]. Whether a PROM-based monitoring and alert system works to improve treatment outcomes for patients after undergoing hip or knee replacements in a cost-effective way has not been tested yet. Particularly because of

the high prevalence of these procedures and the associated healthcare expenditure, even small effects or cost savings can have a considerable impact on system level. Ressources can be used more efficiently, and clinical practice and patient-centeredness can be enhanced through standardised post-surgery follow-ups. Evidence on this field can further give guidance to the discussions about advancing the digitalisation of the health care sector and further initiate research of using PROMs as monitoring and alert system for other conditions such as chronic diseases.

Therefore, we devised and executed the large-scale, two-arm multicentre patient-level randomised controlled trial (RCT) PROMoting Quality [14]. The main objective of the study was to assess the cost-effectiveness of the PROM-based monitoring and alert intervention to proactively identify critical recovery following joint replacement surgery. The early detection aims to improve health outcomes for patients and at the same time reduce costs for the health system, e.g., by reducing the number of post-surgery in- and outpatient visits. In this paper, we are reporting the primary outcome of the PROMoting Quality trial, the intervention's cost-effectiveness.

## Methods

### Ethics statement

The "PROMoting Quality" study was approved by the ethics committee of the Charité—Universitätsmedizin, Berlin (EA4/169/19) (see S1 Appendix for ethics committee protocol). All participants gave written informed consent for data collection and anaylsis at the time of recruitment and could withdraw from the study at any time without stating reasons until the final follow-up.

### Study design and participants

As specified in the study protocol [14], we executed a prospective multicentre two-armed parallel RCT in 9 different hospitals across Germany. Hospitals recruited patients between October 1, 2019 and December 31, 2020. Inclusion criteria for study participation were adult patients (age 18 years or older) undergoing elective primary hip or knee replacements that matched specific predefined procedure codes (see S1 Table). Exclusion criteria were emergency and life-threatening cases, patients classified under the American Society of Anesthesiologists (ASA) categories 4–6 [19] (i.e., patients with a severe life-threatening disease, moribund patients who are not expected to survive without an operation within 24 h, brain-dead patients), and those lacking direct or indirect access to an e-mail account, or the ability to use digital PROMs. Detailed inclusion and exclusion criteria are listed in the published study protocol [14].

The "PROMoting Quality" study was registered with the German Register for Clinical Studies (DRKS) under DRKS00019916. It was conducted in accordance with the Consolidated Standards of Reporting Trials (CONSORT) guidelines [20], ensuring transparency and accuracy in reporting (see S1 CONSORT Checklist).

### Randomisation and masking

Patients were randomised 1:1 to standard care plus PROM-based monitoring and alert intervention or to standard care only. The randomisation took place at hospital discharge and was carried out automatically via the PROM collection and alert software (Heartbeat ONE). For more information on the software solution, see S2 Appendix. To prevent any potential bias in intervention assignment, allocation sequence concealment was employed. Trial personnel and

participants were kept unaware of the allocation sequence until the actual assignment. Beginning with the 1-month follow-up, study nurses could not be blinded due to the nature of the intervention, which required active participation. At no point of the study, participants were informed about their group allocation.

## Procedures

The study employed a PROM-based post-surgery remote monitoring and alert system for patients in the intervention group in addition to the standard of care (for a schematic illustration of the study design, see S1 Fig). PROMs were collected at hospital admission (baseline) and discharge (randomisation), and at 1-, 3-, 6-, and 12-months post-surgery. The intervention occurred in 4 steps: Patients in the intervention group were monitored and evaluated at 1-, 3-, and 6-months post-surgery based on their digitally collected disease-specific and generic PROMs (step 1). An automated alert was triggered to inform a study nurse when PROM scores in the intervention group significantly deteriorated or surpassed a predefined threshold between 2 measurement points (step 2). Subsequently, patients were contacted by the study nurse to consult on their current health status (step 3). If deemed necessary, they were referred to their aftercare physician or respective specialists to discuss possible treatment and medication adjustments, for example, to receive (additional) physiotherapy or adjust pain medication (step 4).

Patients in the control group received the standard of care and, for evaluation purposes, PROM questionnaires at hospital admission, discharge, and 12-months post-surgery. Standard of care in this context includes the clinical patient pathway, i.e., hospital admission, surgery, usually 4 to 7 days of inpatient hospitalisation, and hospital discharge, followed by inpatient or outpatient rehabilitation. After rehabilitation, patients receive aftercare (e.g., check-ups) in a non-standardised manner, usually by their outpatient specialist or general practitioner (GP).

For a detailed description of the study design, see Kuklinski and colleagues [14], and for further information on threshold alerts, see S2 and S3 Tables.

## Outcomes

For the selection of outcome measures, we followed the International Consortium for Health Outcomes Measurement (ICHOM) standard set for Hip- and Knee-Osteoarthritis [21], with minor modifications. We employed a range of PROMs, including the EuroQol 5 dimensions, 5 levels (EQ-5D-5L) and the EuroQol visual analogue scale (EQ-VAS) [22] to capture HRQoL, as well as Hip Disability and Osteoarthritis Outcome Score Physical Function Short-form (HOOS-PS) and Knee Injury and Osteoarthritis Outcome Score Physical Function Short-form (KOOS-PS) [23,24] to assess joint-associated problems and functionality. Analogous pain scales were utilised to evaluate pain in hip (left and right), knee (left and right), and lower back. In addition, Patient-Reported Outcomes Measurement Information System (PROMIS) Depression Shortform (PROMIS-D-SF) and Fatigue Shortform (PROMIS-F-SF) [25] were included to capture patients' mental health status (see S4 Table). Additionally, data on adverse events, such as reoperations and readmissions within 30, 60, and 90 days, were collected.

We defined patient-reported outcomes (PRO) as PROM-score differences between hospital admission and 12-months follow-up. As per study protocol [14], for the effect evaluation of the cost-effectiveness analysis (CEA) we were using a PRO composite measure (PRO-CM) as the main effect parameter, which combines multiple PROs into a multidimensional quality index following Schöner and colleagues [26]. EQ-5D-5L and EQ-VAS formed the generic HRQoL dimension, HOOS-PS/KOOS-PS, and the pain scales formed the physical health dimension, while PROMIS Depression and Fatigue formed the mental health dimension. The

individual PROs were standardised to z-scores and assigned weights following the preference-based approach from Schöner and colleagues [26] before being aggregated via linear additive aggregation. The resulting index was transformed onto a scale with theoretical values ranging from 0 to 100 (T-Score with predefined mean = 50 and standard deviation (SD) = 10), with 0 representing the worst and 100 representing the best possible outcome.

## Cost data

For the CEA, we examined potential cost effects of the intervention on the utilisation and provision of outpatient medical care (including prescription of drugs, remedies such as physical therapy), as well as in- and outpatient hospital treatments, and compared them between study arms over a 1-year period post-surgery. The costs of the intervention (including personnel and software implementation) were also considered in the cost analysis.

We sourced treatment costs from health insurance claims data of 24 participating statutory health insurances, i.e., BARMER and 23 of the BKK Dachverband (i.e., the federal association of company health insurances), which collectively cover approximately 18 million persons in Germany (approx. 23% of the population). Patient-level cost data was available on high detail for the period from 1 year pre- to 1 year post-surgery.

Intervention costs were divided into personnel costs, software implementation, and user license fees. Personnel resources in terms of working minutes were established through structured interviews with the study nurses and priced at the average minutely wage of German care personnel (see S5 and S6 Tables). Costs for software implementation and license fees were derived in close collaboration with the software provider.

To determine incremental costs, we adopted the statutory health insurance (i.e., payer) perspective by combining all treatment costs with intervention costs included in the intervention group.

## Economic and statistical analysis

The PROMoting Quality trial was designed with the assumption of a 0.15 SD change in the main effect parameter PRO-CM, with a 5% error probability and 80% power. This threshold was chosen based on benchmarks in the literature, where 0.2, 0.5, and 0.8 SDs are considered "small," "medium," and "large" effect sizes, respectively [27]. Due to the lack of comparable interventions in existing literature and the expectation of small effects, a conservative 0.15 SD change was assumed to avoid underpowering the study. For more details on the a priori power calculation and statistical analysis plan of the PROMoting Quality trial, see S1 Appendix.

Missing 12-month PROM-scores were imputed with random-forest-based multiple imputation, assuming data was missing at random (no evidence was found that data was missing depending on baseline characteristics or previous score developments).

We performed a comparative analysis using parametric two-sample *t* tests (for normal distributions) and nonparametric Wilcoxon rank-sum tests (for non-normal distributions) at a two-sided 5% significance level to compare baseline characteristics between the 2 study arms.

We employed a mixed effects modelling approach to examine the intervention effect on the PRO-CM, as well as on the individual secondary outcomes EQ-5D-5L, EQ-VAS, HOOS-PS, KOOS-PS, PROMIS-D-SF, PROMIS-F-SF, Pain, and the post-surgery treatment costs. We controlled for age, sex, mobilisation (rapid recovery <6 h after surgery or conventional care otherwise), and body mass index (BMI) as fixed effects, which were selected based on their theoretical relevance and previous literature [28]. Additionally, we incorporated the hospital as a random intercept in the model to account for potential cluster effects on hospital level. In a sensitivity analysis, we calculated regression models in which we also controlled for the

baseline scores of the respective PROMs. Further, in a comparative analysis we employed *t* tests to assess the significance of group differences in the mean of each outcome variable. Statistical significance was judged at the two-sided 5% level and treatment effect estimates are presented with corresponding 95% confidence intervals (CIs).

For the economic evaluation, we applied a within-trial cost-effectiveness approach (i.e., we analyse the costs and outcomes observed during the trial duration without extrapolating data beyond the study period) over the 12-month post-surgery period following intention-to-treat principles. We conducted a generalised CEA based on the Consolidated Health Economic Evaluation Reporting Standards (CHEERS) 2022 [29].

Due to the study duration of 12 months, no discounting of costs or effects was applied. The individual costs were initially analysed descriptively and then examined for differences between the study arms using parametric and nonparametric tests (*t* test, Wilcoxon rank-sum test). Since cost data were right-skewed and long-tailed, with extreme outliers above the 95% percentile, we performed a 95% winsorisation on the cost data (i.e., set all data above the 95% percentile to the 95% percentile) to avoid large distortions of the means and SDs. We found the 95% threshold to be the most appropriate to enable robust statistical analyses without excessively distorting the distribution or sacrificing valuable data points. To adjust for baseline cost differences between study arms, we utilised linear regression to adjust the post-surgery costs for differences in the pre-surgery costs [30].

Outcomes and costs were calculated as mean difference between control and intervention group. These incremental outcomes (nominator) and incremental costs (denominator) were used to calculate the study's primary outcome, i.e., the incremental cost-effectiveness ratio (ICER). The ICER indicates how much additional cost is incurred by an additional unit of outcome.

To assess uncertainty associated with the economic evaluation, we performed a nonparametric bootstrapping (1,000 replications) in the incremental costs and effects. Results are reported as 95% CIs of incremental costs and incremental effects. Data points in the northwest and south-east quadrant of the cost-effectiveness plane represent a dominated or dominant strategy, respectively. Further, we report the probability of dominance, which we define as the proportion of point estimates with negative incremental costs and positive incremental effects (i.e., dominant point estimates in the south-east quadrant of the cost-efectiveness plane).

## Sensitivity and scenario analyses

We focused our analysis and interpretation on 2 main models that only differed in the composition of costs (see Table 1: Base Case A and B). In a sensitivity analysis, we compared these results with 3 alternative scenarios to examine the robustness towards a different outcome composition (Table 1, Alternative 1 and 2) and checked for a potential imputation bias by applying the analysis only for the complete case, i.e., patients that answered all PROM surveys (Table 1, Alternative 3).

Since the application of the PRO-CM as main effect parameter was, to our knowledge, novel and could face challenges in its in interpretability we conducted 2 additional CEAs applying well-known and validated PROMs to test the robustness of the results. Hence, we performed a CEA with the generic EQ-5D-5L as effect parameter to assess the cost-effectiveness in terms of HRQoL and another CEA with the disease-specific HOOS-PS and KOOS-PS, respectively, to assess the cost-effectiveness in terms of physical health. The cost side in these additional robustness analyses were kept equal to the Base case A.

**Table 1. Models for cost-effectiveness analysis and sensitivity checks.**

|  | Outcome weights[a] | Costs[b] | Imputation[c] |
|---|---|---|---|
| Base case A | Differential weights | Treatment costs | Intention-to-treat |
| Base case B | Differential weights | Treatment + intervention costs | Intention-to-treat |
| Alternative 1 | Equal weights | Treatment costs | Intention-to-treat |
| Alternative 2 | Equal weights | Treatment + intervention costs | Intention-to-treat |
| Alternative 3 | Differential weights | Treatment + intervention costs | Complete case |

[a]For the composite index individual indicators were assigned either differential weights with more weight on physical health or equal weights.

[b]Treatment costs are costs from the health insurance claims data that are directly related to the treatment; intervention costs are the additional software and personnel costs associated with the intervention.

[c]Intention-to-treat: Analysis followed intention-to-treat principle and missing data was imputed; complete case: Only complete data was analysed, i.e., of patients who answered all PROM questionnaires.

## Results

### Study population baseline characteristics

Between October 2019 and December 2020, 7,827 patients were recruited from nine hospitals across Germany. After exclusions, the remaining 6,807 patients were randomly assigned to the intervention or control group. Economic evaluation could be conducted on a group of 1,038 patients who belonged to one of the 24 participating health insurances and provided consent for the utilisation of their health insurance claims data. This yields 546 patients with hip replacement (Intervention: 284, Control: 262) and 492 patients with knee replacement (Intervention: 238, Control: 254) for whom we collected sufficient data to assess the primary outcome (see Fig 1). For the complete case analysis, i.e., without multiple imputation, a total of 433 patients with hip replacement (approximately 79%) and 388 patients with knee replacement (approximately 79%) answered all questionnaires.

The baseline characteristics are summarised in Table 2. Patients with hip replacement had a mean age of 66.3 years on the day of surgery, patients with knee replacement 65.7 years. For both procedures, there was a higher proportion of female patients, with 65.4% for patients with hip replacement and 61.8% for patients with knee replacement. Approximately 32.8% of patients with hip replacement and 53.1% of patients with knee replacement were obese (i.e., BMI $\geq$ 30), and 38.4% of the patients with hip replacement, but only 29.0% of the patients with knee replacement had no comorbidities recorded. All baseline characteristics were also tested for differences between the groups, and we found no statistically significant differences.

### Outcomes

Comparative analyses showed that, for patients with hip replacement, the PRO-CM [26] was significantly higher (2.54 incremental points; 95% CI [0.87, 4.21]; $t$ test $p = 0.003$) in the intervention group, namely with 51.22 (SD 9.66), than for patients in the control group, namely with 48.68 (SD 10.21). The intervention group thus had a 5.22% higher PRO-CM than the control group (see Table 3).

These findings were supported in the regression results: Fig 2 visualises the mixed effect model point estimates of the intervention effect on PRO-CM, on secondary outcomes, and on post-surgery costs. It shows that the intervention had a highly significant impact on the effect measure and increased the PRO-CM on average by 0.24 SD (95% CI [0.07, 0.40]; $p = 0.005$).

Breaking it down to the individual PROs, we found significant differences among patients with hip replacement between the intervention and the control group for all PROs except for

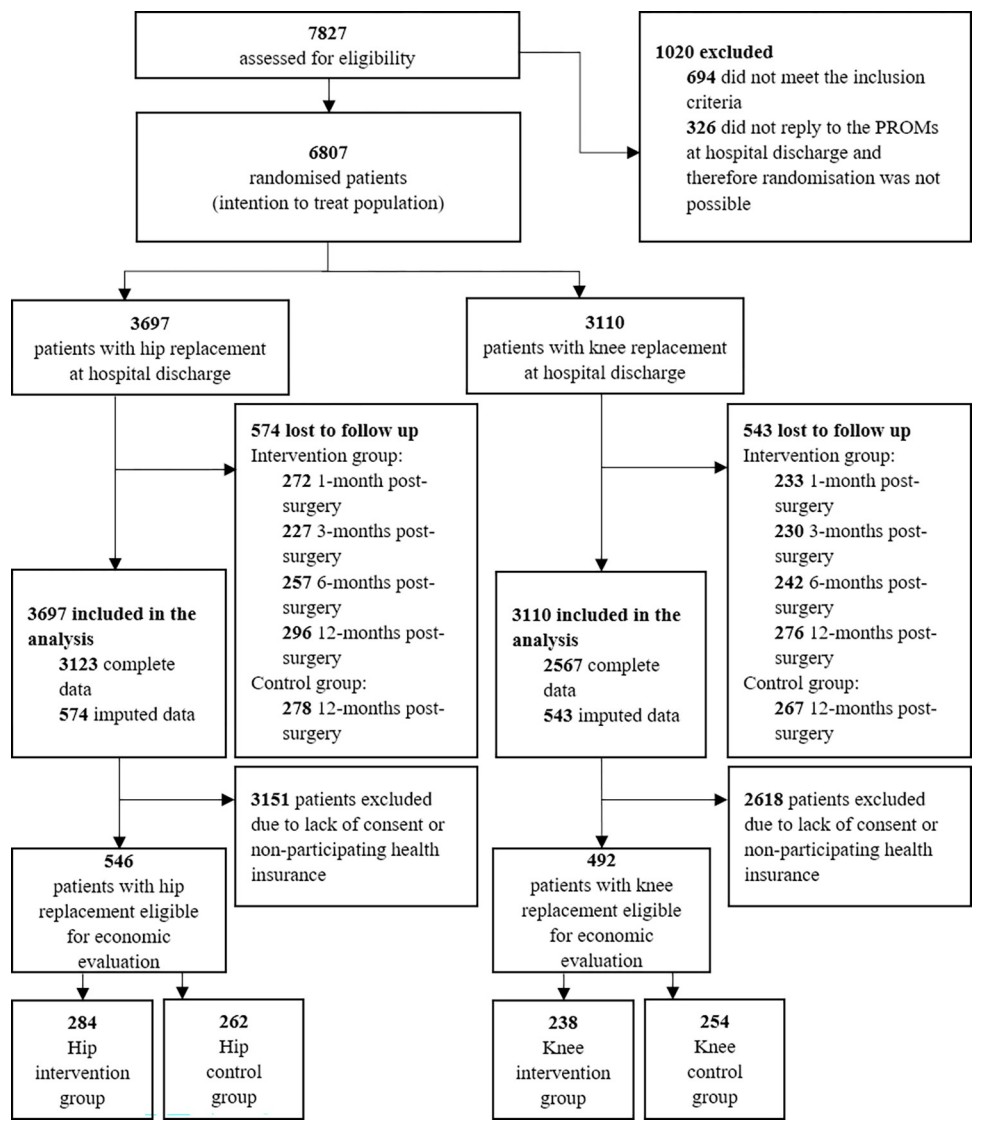

**Fig 1. Trial profile.** PROMs, patient-reported outcome measures.

fatigue symptoms and pain. The largest score improvements were observed in the physical health dimensions HOOS-PS and pain scores, with the intervention group showing a higher improvement in HOOS-PS with 37.38 (SD 17.23) points compared to the control group with 32.76 (SD 19.09) points (4.62 incremental points; 95% CI [1.55, 7.67]; $t$ test $p$ = 0.003). The point estimates in Fig 2 also show that the intervention effect size was largest for HOOS-PS improvement. The second highest improvement was observed in the HRQoL dimension measured by EQ-VAS. The intervention group had a significantly higher mean change of 23.46 (SD 20.92) compared to the control group with 18.64 (SD 23.10) (4.82 incremental points; 95% CI [1.05, 8.59]; $t$ test $p$ = 0.012). EQ-5D-5L improved on average by 0.336 (SD 0.263) for patients who received the PROM intervention compared to 0.277 (SD 0.278) for patients in control group (0.059 incremental points; 95% CI [0.014, 0.105]; $t$ test $p$ = 0.011). Lastly, patients with hip replacement in the intervention group had a mean improvement in depressive symptoms, captured by PROMIS-D-SF, of 3.41 (SD 7.44) compared to 1.87 (SD 8.46) in the control group (1.53 incremental points; 95% CI [0.20, 2.87]; $t$ test $p$ = 0.025).

**Table 2. Descriptive statistics of the study population.**

| | Patients with hip replacement | | | Patients with knee replacement | | |
|---|---|---|---|---|---|---|
| | Intervention | Control | Total | Intervention | Control | Total |
| | (N = 284) | (N = 262) | (N = 546) | (N = 238) | (N = 254) | (N = 492) |
| **Age** | | | | | | |
| mean (SD) | 65.7 (10.6) | 66.9 (10.2) | 66.3 (10.4) | 66.0 (8.8) | 65.5 (9.8) | 65.7 (9.3) |
| **Sex (%)** | | | | | | |
| Female | 186 (65.5) | 171 (65.3) | 357 (65.4) | 149 (62.6) | 155 (61.0) | 304 (61.8) |
| Male | 98 (34.5) | 91 (34.7) | 189 (34.6) | 89 (37.4) | 99 (39.0) | 188 (38.2) |
| **BMI (%)** | | | | | | |
| Underweight | 2 (0.7) | 2 (0.8) | 4 (0.7) | 0 (0.0) | 0 (0.0) | 0 (0.0) |
| Normal | 84 (29.6) | 82 (31.3) | 166 (30.4) | 33 (13.9) | 31 (12.2) | 64 (12.0) |
| Overweight | 111 (39.1) | 86 (32.8) | 197 (36.1) | 79 (33.2) | 88 (34.7) | 167 (33.9) |
| Obese | 87 (30.6) | 92 (35.1) | 179 (32.8) | 126 (52.9) | 135 (53.2) | 261 (53.1) |
| **Current smoker (%)** | | | | | | |
| No | 232 (81.7) | 225 (85.9) | 457 (83.7) | 204 (85.7) | 223 (87.8) | 427 (86.8) |
| Yes | 52 (18.3) | 37 (14.1) | 89 (16.3) | 34 (14.3) | 31 (12.2) | 65 (13.2) |
| **Education (%)** | | | | | | |
| no school degree | 2 (0.7) | 1 (0.4) | 3 (0.5) | 3 (1.3) | 2 (0.8) | 5 (1.0) |
| primary school degree | 29 (10.2) | 30 (11.5) | 59 (10.8) | 44 (18.5) | 36 (14.2) | 80 (16.3) |
| high/middle school degree | 202 (71.1) | 176 (67.2) | 378 (69.2) | 154 (64.7) | 170 (66.9) | 324 (65.9) |
| university degree | 51 (18.0) | 55 (21.0) | 106 (19.4) | 37 (15.6) | 46 (18.1) | 83 (16.9) |
| **Living situation (%)** | | | | | | |
| Alone | 59 (20.8) | 66 (25.2) | 125 (22.9) | 48 (20.2) | 58 (22.8) | 106 (21.5) |
| care facility | 1 (0.4) | 0 (0.0) | 1 (0.2) | 0 (0) | 2 (0.8) | 2 (0.4) |
| with partner/family/friends | 222 (78.2) | 195 (74.4) | 417 (76.4) | 190 (79.8) | 190 (74.8) | 380 (77.2) |
| Other | 2 (0.7) | 1 (0.4) | 3 (0.5) | 0 (0) | 4 (1.6) | 4 (0.8) |
| **Employment status (%)** | | | | | | |
| looking for work | 4 (1.4) | 3 (1.1) | 7 (1.3) | 0 (0.0) | 1 (0.4) | 1 (0.2) |
| unable to work due to arthrosis | 19 (6.7) | 18 (6.9) | 37 (6.8) | 20 (8.4) | 18 (7.1) | 38 (7.7) |
| unable to work due to other disease | 7 (2.5) | 7 (2.7) | 14 (2.6) | 5 (2.1) | 11 (4.3) | 16 (3.3) |
| working PT | 32 (11.3) | 19 (7.3) | 51 (9.3) | 34 (14.3) | 23 (9.1) | 57 (11.6) |
| working FT | 54 (19.0) | 56 (21.4) | 110 (20.1) | 43 (18.1) | 44 (17.3) | 87 (17.7) |
| voluntarily not working | 168 (59.2) | 159 (60.7) | 327 (59.9) | 136 (57.1) | 157 (61.8) | 293 (59.6) |
| **Job-related physical activity (%)** | | | | | | |
| not assessable | 27 (9.5) | 20 (7.6) | 47 (8.6) | 136 (57.1) | 157 (61.8) | 293 (59.6) |
| predominantly sedentary activities | 113 (39.8) | 117 (44.7) | 230 (42.1) | 22 (9.2) | 20 (7.9) | 42 (8.5) |
| light physical activities | 51 (18.0) | 53 (20.2) | 104 (19.0) | 80 (33.6) | 94 (37.0) | 174 (35.4) |
| moderate physical activities | 61 (21.5) | 48 (18.3) | 109 (20.0) | 41 (17.2) | 51 (20.1) | 92 (18.7) |
| heavy physical activities | 32 (11.3) | 24 (9.2) | 56 (10.3) | 62 (26.1) | 60 (23.6) | 122 (24.8) |
| **Nursing care level (%)** | | | | | | |
| None | 275 (96.8) | 254 (96.9) | 529 (96.9) | 231 (97.1) | 241 (94.9) | 472 (95.9) |
| level 1 | 4 (1.4) | 4 (1.5) | 8 (1.5) | 2 (0.8) | 4 (1.6) | 6 (1.2) |
| level 2 | 3 (1.1) | 3 (1.1) | 6 (1.1) | 3 (1.3) | 9 (3.5) | 12 (2.4) |
| level 3 | 2 (0.7) | 1 (0.4) | 3 (0.5) | 2 (0.8) | 0 (0.0) | 2 (0.4) |
| **Mobilisation after surgery (%)** | | | | | | |
| within 6 hours (RR) | 135 (47.5) | 138 (52.7) | 273 (50.0) | 124 (52.1) | 109 (42.9) | 233 (47.4) |
| within 12 hours (CC) | 81 (28.5) | 70 (26.7) | 151 (27.7) | 72 (30.3) | 62 (24.4) | 134 (27.2) |
| within 24 hours (CC) | 61 (21.5) | 49 (18.7) | 110 (20.2) | 34 (14.3) | 74 (29.1) | 108 (22.0) |

*(Continued)*

**Table 2.** (Continued)

| | Patients with hip replacement | | | Patients with knee replacement | | |
|---|---|---|---|---|---|---|
| | **Intervention** | **Control** | **Total** | **Intervention** | **Control** | **Total** |
| | ($N = 284$) | ($N = 262$) | ($N = 546$) | ($N = 238$) | ($N = 254$) | ($N = 492$) |
| within 48 hours (CC) | 2 (0.7) | 3 (1.2) | 5 (0.9) | 6 (2.5) | 6 (2.4) | 12 (2.4) |
| after 48 hours (CC) | 5 (1.8) | 2 (0.8) | 7 (1.3) | 2 (0.8) | 3 (1.2) | 5 (1.0) |
| **Daily physical activity (%)** | | | | | | |
| None | 31 (10.9) | 25 (9.5) | 56 (10.3) | 33 (13.9) | 29 (11.4) | 62 (12.6) |
| up to 30 min | 32 (11.3) | 21 (8.0) | 53 (9.7) | 21 (8.8) | 27 (10.6) | 48 (9.8) |
| up to 1h | 36 (12.7) | 33 (12.6) | 69 (12.6) | 23 (9.7) | 19 (7.5) | 42 (8.5) |
| up to 2 h | 47 (16.5) | 49 (18.7) | 96 (17.6) | 32 (13.4) | 31 (12.2) | 63 (12.8) |
| more than 2 h | 138 (48.6) | 134 (51.1) | 272 (49.8) | 52 (21.8) | 41 (16.1) | 93 (18.9) |
| **Comorbidities (%)** | | | | | | |
| None | 109 (38.4) | 94 (35.9) | 203 (37.2) | 69 (29.0) | 61 (24.0) | 130 (26.4) |
| Arthrosis/arthritis | 51 (18.0) | 48 (18.3) | 99 (18.1) | 49 (20.6) | 46 (18.1) | 95 (19.3) |
| Cancer (within last 5 years) | 11 (3.9) | 16 (6.1) | 27 (4.9) | 16 (6.7) | 16 (6.3) | 32 (6.5) |
| Circulation-disturbances | 6 (2.1) | 10 (3.8) | 16 (2.9) | 11 (4.6) | 10 (3.9) | 21 (4.3) |
| Depression | 16 (5.6) | 19 (7.3) | 35 (6.4) | 17 (7.1) | 23 (9.1) | 40 (8.1) |
| Diabetes mellitus | 25 (8.8) | 26 (9.9) | 51 (9.3) | 25 (10.5) | 35 (13.8) | 60 (12.2) |
| Diseases of the nervous system | 4 (1.4) | 4 (1.5) | 8 (1.5) | 6 (2.5) | 3 (1.2) | 9 (1.8) |
| Heart disease | 38 (13.4) | 32 (12.2) | 70 (12.8) | 32 (13.4) | 39 (15.4) | 71 (14.4) |
| Hypertension | 128 (45.1) | 136 (51.9) | 264 (48.4) | 143 (60.1) | 157 (61.8) | 300 (61.0) |
| Kidney disease | 8 (2.8) | 10 (3.8) | 18 (3.3) | 6 (2.5) | 6 (2.4) | 12 (2.4) |
| Liver disease | 5 (1.8) | 4 (1.5) | 9 (1.6) | 6 (2.5) | 4 (1.6) | 10 (2.0) |
| Lung disease | 26 (9.2) | 21 (8.0) | 47 (8.6) | 23 (9.7) | 23 (9.1) | 46 (9.3) |
| Rheumatoid arthritis | 11 (3.9) | 13 (5.0) | 24 (4.4) | 14 (5.9) | 29 (11.4) | 43 (8.7) |
| Stroke-related disabilities | 7 (2.5) | 8 (3.1) | 15 (2.7) | 5 (2.1) | 7 (2.8) | 12 (2.4) |
| **PROM baseline score means (SD)** | | | | | | |
| EQ-5D-5L | 0.564 (0.263) | 0.599 (0.246) | 0.581 (0.255) | 0.615 (0.235) | 0.606 (0.253) | 0.610 (0.244) |
| EQ-VAS | 54.0 (18.5) | 55.1 (19.2) | 54.5 (18.8) | 56.8 (18.5) | 57.9 (18.5) | 57.3 (18.5) |
| HOOS-/KOOS-PS | 50.3 (16.4) | 47.7 (15.3) | 49.0 (16.0) | 42.6 (11.5) | 43.4 (10.9) | 43.0 (11.2) |
| PROMIS-F-SF | 48.8 (9.9) | 48.5 (10.6) | 48.6 (10.2) | 47.9 (9.8) | 48.2 (9.9) | 48.0 (9.9) |
| PROMIS-D-SF | 50.2 (8.4) | 49.7 (8.2) | 49.9 (8.3) | 49.4 (8.5) | 49.7 (8.6) | 49.6 (8.6) |
| Pain Score | 2.9 (1.5) | 2.9 (1.3) | 2.9 (1.4) | 2.8 (1.3) | 2.8 (1.3) | 2.8 (1.3) |

BMI, body mass index; (FT), full time; (PT), part time; (RR), rapid recovery, i.e., mobilisation within 6 h post-surgery; (CC), conventional care, i.e., mobilisation >6 h post-surgery; PROM, patient-reported outcome measures; SD, standard deviation; EQ-5D-5L, EuroQol 5 dimensions, 5 levels; EQ-VAS, EuroQol virtual analogue scale; HOOS-PS, Hip Disability and Osteoarthritis Outcome Score Physical Function Short-form; KOOS-PS, Knee Injury and Osteoarthritis Outcome Score Physical Function Short-form; PROMIS, Patient-Reported Outcomes Measurement Information System Depression Shortform (PROMIS-D-SF) and Fatigue Shortform (PROMIS-F-SF).

For patients with knee replacement, the PRO-CM showed a higher score of 50.45 (SD 9.87) for the intervention group compared to the control group with 49.58 (SD 10.12). This PRO-CM difference of 1.8% between groups was not significant (Table 3). However, we found significant differences in individual score improvements of EQ-VAS and PROMIS-D-SF.

The improvement of the intervention group in EQ-VAS was 17.07 (SD 21.11) compared to 13.08 (SD 22.78) in the control group (3.99 incremental points; 95% CI [0.10, 7.89]; $t$ test $p = 0.045$). PROMIS-D-SF improvement in the intervention was 2.75 (SD 8.13), which is significantly different from the control group with 1.19 (SD 8.12) (1.56 incremental points; 95%

**Table 3. Incremental effects and costs—Results of the comparative analysis.**

| Patients with hip replacement | | | | | |
|---|---|---|---|---|---|
| | **Total** | **Intervention group** | **Control group** | **Incremental[a] [95% CI]** | **p-value[b]** |
| **PROs–mean (SD)** | | | | | |
| PRO-CM | 50.00 (10.00) | 51.22 (9.66) | 48.68 (10.21) | 2.54 [0.87, 4.21] | $p = 0.003^{(t)}$ |
| EQ-5D-5L | 0.308 (0.217) | 0.336 (0.263) | 0.277 (0.278) | 0.059 [0.014, 0.105] | $p = 0.011^{(t)}$ |
| EQ-VAS | 21.14 (22.51) | 23.46 (20.92) | 18.64 (23.10) | 4.82 [1.05, 8.59] | $p = 0.012^{(t)}$ |
| HOOS-PS | 35.16 (18.27) | 37.38 (17.23) | 32.76 (19.09) | 4.62 [1.55, 7.67] | $p = 0.003^{(t)}$ |
| PROMIS-D-SF | 2.67 (7.98) | 3.41 (7.44) | 1.87 (8.46) | 1.53 [0.20, 2.87] | $p = 0.025^{(t)}$ |
| PROMIS-F-SF | 3.69 (9.63) | 4.09 (9.42) | 3.26 (9.84) | 0.83 [−0.79, 2.45] | Not sign. |
| Pain Score | 1.81 (1.52) | 1.85 (1.56) | 1.77 (1.49) | 0.08 [−0.18, 0.33] | Not sign. |
| **Adverse events–n (%)** | | | | | |
| 1-year Reoperation | | 2 (0.7) | 6 (2.3) | 4 | Not sign. |
| 1-year Readmission | | 4 (1.4) | 6 (2.3) | 2 | Not sign. |
| **Treatment costs (€)–mean (SD)** | | | | | |
| Pre-surgery | 4,186.60 (7,712.58) | 3,615.90 (4,833.44) | 4,805.23 (9,906.55) | −1,189.33 [−2,484.43, 105.78] | $p = 0.025^{(w)}$ |
| Post-surgery | 4,804.95 (6,648.23) | 4,226.26 (5,575.47) | 5,432.22 (7,604.43) | −1,205.96 [−2,321.06, −90.86] | $p = 0.019^{(w)}$ |
| Adjusted post-surgery | 3,800.75 (1,608.78) | 3,620.12 (1,544.80) | 3,996.55 (1,656.16) | −376.43 [−645.52, −107.33] | $p = 0.004^{(w)}$ |
| **Intervention costs per capita (€)** | | | | | |
| Personnel | | 12.39 | NA | | |
| Implementation | | 11.39 | NA | | |
| License fees | | 100.00 | NA | | |
| Total | | 123.78 | NA | | |

| Patients with knee replacement | | | | | |
|---|---|---|---|---|---|
| | **Total** | **Intervention group** | **Control group** | **Incremental[a] [95% CI]** | **p-value[b]** |
| **PROs–mean (SD)** | | | | | |
| PRO-CM | 50.00 (10.00) | 50.45 (9.87) | 49.58 (10.12) | 0.87 [−0.91, 2.64] | Not sign. |
| EQ-5D-5L | 0.240 (0.263) | 0.249 (0.256) | 0.231 (0.271) | 0.018 [−0.03, 0.06] | Not sign. |
| EQ-VAS | 15.01 (22.06) | 17.07 (21.11) | 13.08 (22.78) | 3.99 [0.10, 7.89] | $p = 0.045^{(t)}$ |
| KOOS-PS | 17.52 (14.08) | 17.51 (14.44) | 17.53 (13.76) | −0.01 [−2.52, 2.48] | Not sign. |
| PROMIS-D-SF | 1.95 (8.20) | 2.75 (8.13) | 1.19 (8.21) | 1.56 [0.11, 3.01] | $p = 0.035^{(t)}$ |
| PROMIS-F-SF | 2.54 (9.06) | 3.22 (9.27) | 1.90 (8.81) | 1.32 [−0.28, 2.92] | Not sign. |
| Pain Score | 1.40 (1.51) | 1.39 (1.44) | 1.42 (1.57) | −0.03 [−0.30, 0.24] | Not sign. |
| **Adverse events—n (%)** | | | | | |
| 1-year Reoperation | | 8 (3.4) | 11 (4.3) | 3 | Not sign. |

*(Continued)*

**Table 3.** (Continued)

| | | | | | |
|---|---|---|---|---|---|
| 1-year Readmission | | 8 (3.4) | 9 (3.5) | 2 | Not sign. |
| **Treatment costs (€)–mean (SD)** | | | | | |
| Pre-surgery | 4,989.97 (6,190.03) | 4,221.54 (4,898.17) | 5,710.00 (7,128.14) | −1,488.45 [−2,578.81, 398.09] | $p = 0.010^{(w)}$ |
| Post-surgery | 6,011.78 (7,523.84) | 5,298.20 (6,041.66) | 6,680.41 (8,645.18) | −1,382.20 [−2,711.56, −52.85] | $p = 0.073^{(w)}$ |
| Adjusted post-surgery | 5,101.48 (2,236.39) | 4,907.62 (2,186.07) | 5,283.13 (2,271.84) | −375.50 [−770.91, 19.91] | $p = 0.034^{(w)}$ |
| **Intervention costs per capita (€)** | | | | | |
| Personnel | | 12.39 | NA | | |
| Implementation | | 11.39 | NA | | |
| License fees | | 100.00 | NA | | |
| Total | | 123.78 | NA | | |

PROs, patient-reported outcomes; SD, standard deviation; CI, confidence interval; EQ-5D-5L, EuroQol 5 dimensions, 5 levels; EQ-VAS, EuroQol virtual analogue scale; HOOS-PS, Hip Disability and Osteoarthritis Outcome Score Physical Function Short-form; KOOS-PS, Knee Injury and Osteoarthritis Outcome Score Physical Function Short-form; PROMIS, Patient-Reported Outcomes Measurement Information System Depression Shortform (PROMIS-D-SF) and Fatigue Shortform (PROMIS-F-SF); Not sign., not significant; NA, not applicable.

[a]Incremental is calculated as effect(intervention group)—effect(control group).

[b](t) Indicate results from *t* tests, (w) results from Wilcoxon rank-sum tests.

CI [0.11, 3.01]; *t* test *p* = 0.035). The displayed point estimates of the mixed effects model in Fig 2 show similar patterns.

Results of the sensitivity analyses that additionally controlled for baseline PROM scores show similar patterns in terms of significant mean differences while the effect size slightly decreased for patients with hip replacement. In patients with knee replacement controlling for the baseline scores increased the effect sizes albeit only slightly, while it showed a significant

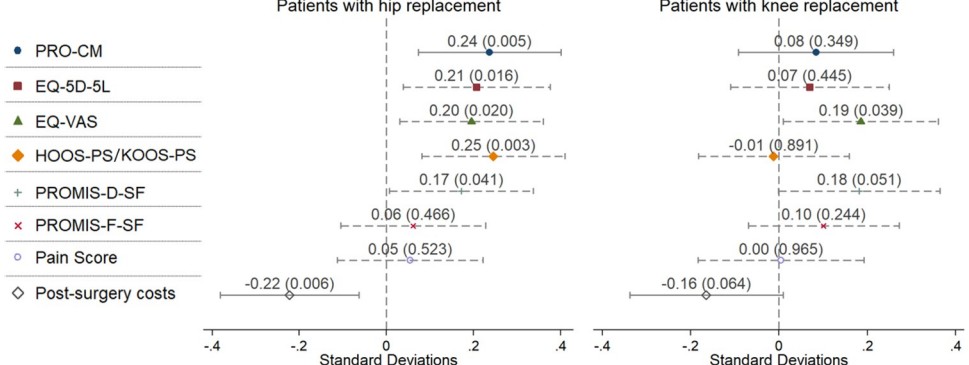

**Fig 2. Mixed effect model estimates for the intervention group.** Fig 2 displays the mixed effect model point estimates of the intervention effect on PRO-CM and post-surgery costs (solid lines) as well as on secondary outcomes (dashed lines). The lines represent their corresponding 95% CIs. The numbers in brackets indicate the corresponding *p*-values. Outcomes and costs were standardised to z-scores to enable comparability. Point estimates above zero indicate better health outcomes or higher costs in the intervention group (in SDs); point estimates below zero indicate worse health outcomes or lower costs in the intervention group compared to the control group. In patients with hip replacement HOOS-PS was used, in patients with knee replacement KOOS-PS was used. PRO-CM, patient-reported outcome composite measure; EQ-5D-5L, EuroQol 5 dimensions, 5 levels; EQ-VAS, EuroQol virtual analogue scale; HOOS-PS, Hip Disability and Osteoarthritis Outcome Score Physical Function Short-form; KOOS-PS, Knee Injury and Osteoarthritis Outcome Score Physical Function Short-form; PROMIS, Patient-Reported Outcomes Measurement Information System Depression Shortform (PROMIS-D-SF) and Fatigue Shortform (PROMIS-F-SF).

effect on PROMIS-D-SF. The results of the estimates can be found in S2 Fig for patients with hip replacement and S3 Fig for patients with knee replacement.

Within 1-year post-surgery adverse events ratios (readmission and reoperation) were generally low. Only 4 (1.4%) patients with hip replacement from the intervention group were readmitted to the hospital of which 2 (0.7%) were reoperated. From the control group, 6 patients (2.3%) were readmitted and 6 (2.3%) reoperated. The intervention group of patients with knee replacement had 8 (3.4%) readmissions and 8 (3.4%) reoperations compared to 9 (3.5%) readmissions and 11 (4.3%) reoperations in the control group. Despite the small effect in favour of the intervention group, these differences in adverse events were nonsignificant due to the small number of observations.

## Costs

The adjusted 1-year post-surgery treatment costs were significantly lower in the intervention groups in both patients with hip and patients with knee replacement (Table 3). The intervention group of patients with hip replacement had mean post-surgery costs of 3,620.12€ compared to 3,996.55€ in the control group. This is 376.43€ less in the intervention group (95% CI [−645.52, −−107.33]; Wilcoxon rank-sum test $p = 0.004$). The intervention group of patients with knee replacement had mean post-op costs of 4,907.62€ while the control group had mean costs of 5,283.13€. This is 375.50€ less in the intervention group (95% CI [−770.91, 19.91]; Wilcoxon rank-sum test $p = 0.034$). The mixed effect model, however, only showed a nonsignificant intervention effect on the costs in patients with knee replacement (Fig 2). See S7 and S8 Tables for more details on costs.

## Primary cost-effectiveness analysis

The bootstrapped within-trial CEA for patients with hip replacement showed that during the 12-month follow-up period, the intervention group had larger outcome improvements (2.54 additional PRO-CM units, 95% CI [0.93, 4.14]; $p = 0.002$) at lower cost compared to the control group (−376.43€, 95% CI [−639.74, −113.12]; $p = 0.005$) (see Table 4).

Panel A of Fig 3 shows the results of the bootstrapped point estimates of incremental costs and effects for patients with hip replacement. The observed negative incremental costs and positive incremental effects revealed a dominant ICER. The intervention's probability of dominance (i.e., the proportion of point estimates in the south-east quadrant) was 99.6%. The bootstrapped estimates on panel B of Fig 3 show that additionally considering the intervention costs of 123.78€ per patient, the intervention was still less costly (−252.73€, 95% CI [−516.05, 10.58]; $p = 0.060$) revealing a dominant ICER and a 97.5% probability of dominance.

Regarding patients with knee replacement, the PROM intervention showed to be effective, albeit non-significantly (0.87 additional PRO-CM units, 95% CI [−0.94, 2.67]; $p = 0.347$). At the same time, we observed lower costs in the intervention group (−375.50€, 95% CI [−767.40, 16.39]; $p = 0.060$) (see Table 4). Panel C of Fig 3 shows the results of the bootstrapped point estimates of incremental costs and incremental effects for patients with knee replacement. Compared to patients with hip replacement, we found visibly less point estimates in the southeast (i.e., dominant) quadrant in patients with knee replacement where we also observed some estimates in the south-west quadrant, in which effects are negative but costs are reduced. However, the probability of dominance was still 79.9%. Panel D of Fig 3 shows that adding the intervention costs to the treatment costs shifted the estimates cloud slightly up. It resulted in a cost difference between intervention and control group of −251.81€ (95% CI [−643.71, 140.09]; $p = 0.208$). The probability of dominance was 73.3%.

Since the intervention revealed dominant ICERs for both procedures, we do not report the exact ICER values as negative ratios are difficult to interpret and can be misleading.

**Table 4. Incremental costs, effects, and ICER.**

| Hip replacement patients | | | | | |
|---|---|---|---|---|---|
| | **Incremental costs** | | **Incremental PRO-CM** | | **ICER** |
| | **Mean (SE)** | **CI** | **Mean (SE)** | **CI** | |
| Base Case A[a] | −376.43 (134.35) | [−639.74; −113.12] | 2.54 (0.82) | [0.93; 4.14] | −148.42 |
| Base Case B[b] | −252.73 (134.35) | [−516.05; 10.58] | 2.54 (0.82) | [0.93; 4.14] | −99.65 |
| Sensitivity analyses | | | | | |
| Alternative 1[c] | −376.43 (134.35) | [−639.74; −113.12] | 2.43 (0.82) | [0.82; 4.04] | −154.80 |
| Alternative 2[d] | −252.73 (134.35) | [−516.05; 10.58] | 2.43 (0.82) | [0.82; 4.04] | −103.93 |
| Alternative 3[e] | −317.30 (155.95) | [−622.97; −11.64] | 3.06 (1.02) | [1.05; 5.06] | −103.79 |

| Knee replacement patients | | | | | |
|---|---|---|---|---|---|
| | **Incremental costs** | | **Incremental PRO-CM** | | **ICER** |
| | **Mean (SE)** | **CI** | **Mean (SE)** | **CI** | |
| Base Case A[a] | −375.50 (199.95) | [−767.40; 16.39] | 0.87 (0.92) | [−0.94; 2.67] | −433.71 |
| Base Case B[b] | −251.81 (199.95) | [−643.71; 140.09] | 0.87 (0.92) | [−0.94; 2.67] | −290.84 |
| Sensitivity analyses | | | | | |
| Alternative 1[c] | −375.50 (199.95) | [−767.40; 16.39] | 1.36 (0.92) | [−0.44; 3.16] | −275.44 |
| Alternative 2[d] | −251.81 (199.95) | [−643.71; 140.09] | 1.36 (0.92) | [−0.44; 3.16] | −184.71 |
| Alternative 3[e] | −324.53 (236.87) | [−788.80; 139.73] | 1.48 (1.01) | [−0.50; 3.47] | −219.22 |

[a]Base Case A–PRO-CM with differential weights, only treatment costs, intention-to-treat principle.

[b]Base Case B–PRO-CM with differential weights, treatment and intervention costs, intention-to-treat principle.

[c]Alternative 1 –PRO-CM with equal weights; only treatment costs; intention-to-treat principle.

[d]Alternative 2 –PRO-CM with equal weights; treatment and intervention costs; intention-to-treat principle.

[e]Alternative 3 –PRO-CM with differential weights; treatment and intervention costs; only complete cases, i.e., patients that answered all PROM questionnaires.

PRO-CM, patient-reported outcome composite measure; ICER, incremental cost-effectiveness ratio; SE, standard error; CI, 95% confidence interval.

## Sensitivity analyses

The findings from our base case models were supported by the sensitivity analyses. While a composite measure with equally weighted PROs (Table 1, Alternative 1 and 2) led to very similar mean incremental outcomes of 2.43 (95% CI [0.82, 4.04]; $p = 0.003$) points in the intervention group of patients with hip replacement, the incremental outcomes in the intervention group of patients with knee replacement increased to 1.36 (95% CI [−0.44, 3.16]; $p = 0.138$) (see Table 4).

In the complete case analysis (Table 1, Alternative 3), we only considered patients of both groups that answered the complete set of PROM questionnaires to check for a potential imputation bias. In this scenario, we observed less costs in both intervention groups compared to the control groups, while the incremental outcomes with 3.06 (95% CI [1.05, 5.06]; $p = 0.003$) in patients with hip replacement and 1.48 (95% CI [−0.50, 3.47]; $p = 0.144$) in patients with knee replacement were higher than in all other scenarios. These scenario analyses indicated robust results in terms of larger outcome improvements and cost savings for both intervention groups.

Finally, the 2 CEAs using the EQ-5D-5L and the HOOS-PS and KOOS-PS, respectively, were in line with our base case analysis and further confirmed the robustnesss of the results. For the results of these additional analyses, see S4 and S5 Figs for patients with hip replacement and S6 and S7 Figs for patients with knee replacement. For patients with hip replacement, both the EQ-5D-5L CEA and the HOOS-PS CEA revealed a dominant intervention with

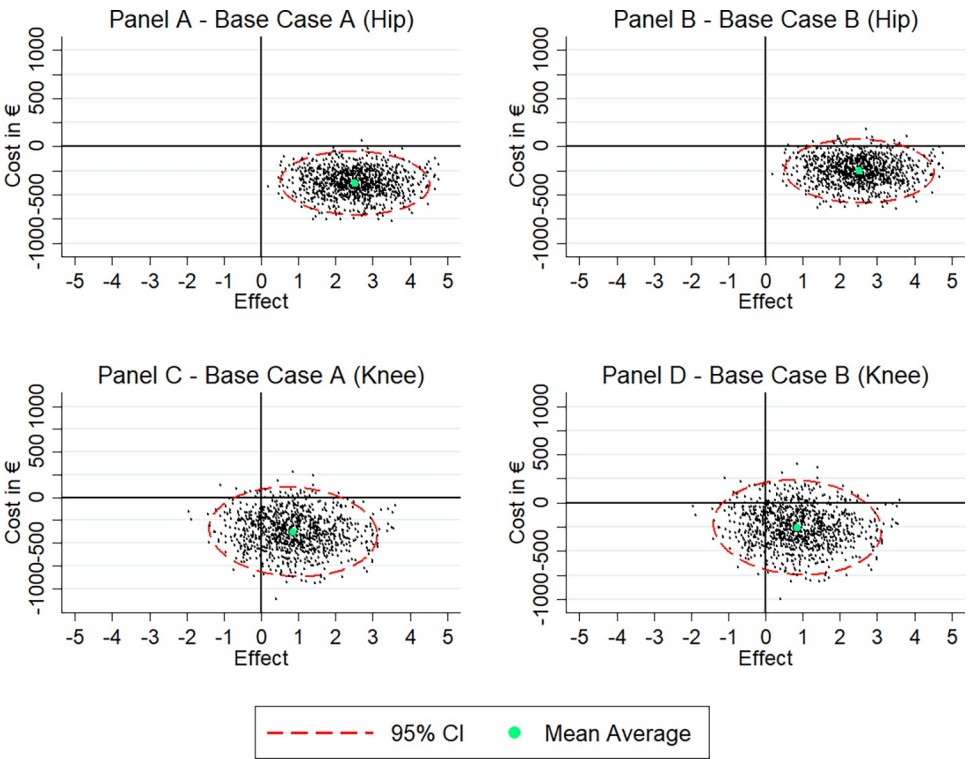

**Fig 3. Cost-effectiveness of the PROM intervention in patients with hip and patients with knee replacement.**

99.6% probability of dominance and 99.8%, respectively. Thereby patients with hip replacement in the intervention group on average improved by 0.22 SD in EQ-5D-5L (95% CI [0.05, 0.39]; $p = 0.011$) and by 0.26 SD in HOOS-PS (95% CI [0.10, 0.42]; $p = 0.002$) more compared to the control group. In patients with knee replacement, the intervention showed less cost-effective, in line with the base case results. The incremental effect in EQ-5D-5L was 0.07 SD (95% CI [−0.12, 0.26]; $p = 0.455$) for the intervention group and in KOOS-PS 0.00 SD (95% CI [−0.18, 0.17]; $p = 0.991$). The probability of dominance was 74.2% and 45.5%, respectively.

## Discussion

We conducted the "PROMoting Quality" trial to explore the cost-effectiveness of a PROM-based monitoring and alert intervention for hip and knee replacement patients within the first 12-months after surgery. Overall, we found the intervention to be cost-effective for both procedures, but to a different degree in size and significance. For patients with hip replacement, the probability of dominance of the intervention was 99.6% without intervention cost (97.5% with intervention cost) with a positive outcome effect of 2.54 additional PRO-CM units and lower cost of −376.43€ (−252.73€) between intervention and control group. These estimations position the intervention in the dominant quadrant of the bootstrapped ICER results, indicating superior outcomes at lower costs. Sensitivity analyses confirmed the robustness of our results.

When examining the hip replacement results more closely, they showed that effects were driven by relatively large improvements of the intervention compared to the control group in nearly all PROM dimensions, except the pain scale and PROMIS-F-SF (see Fig 2), with the largest effects in HRQoL and functional status. Comparing the results to those of all 6,807 patients (i.e., including patients without cost data), Steinbeck and colleagues [31] reveal slight

differences in the affected health dimensions: Main effects of the intervention for all patients were observed in the EQ-VAS, the PROMIS-F-SF, and PROMIS-D-SF, with nonsignificant, smaller effects for functional status measured by HOOS-PS. While both the original sample with all patients and our patient group showed positive effects of the intervention, the differences in the PRO dimensions must be due to slightly different sample characteristics (see S9 Table). In particular, we see that there are differences in the sex distribution: In our patient group, the proportion of female patients was higher in both hip patients (65%) and knee patients (62%) than in the overall sample with all patients (hip 56%, knee 54%), suggesting that female patients might benefit more from the intervention regarding their physical wellbeing which is supported by Langenberger and colleagues [32]. A recent study by Steinbeck and colleagues [33] suggests that future studies should focus more on the gender health gap to identify gender-specific differences and improve patient-centred care. However, apart from this we only see nonsignificant differences in the sample characteristics.

We initially expected to see the major effects for functional status improvements due to treatment protocol adjustments based on PROM-alerts, but also observed significant positive effects for HRQoL and mental health in our sample. This could be caused by the "being taken care of" aspect of remotely monitoring patients via PROMs [34]. Even though we cannot compare our results to similar studies for patients with hip or knee replacements, as to our knowledge they do not exist yet, the improvements in HRQoL of our intervention are in line with studies in other fields [35,36]. In particular for oncologic patients, Basch and colleagues [16], for example, reported a significant positive effect of symptom monitoring via PROMs on the EQ-5D-5L. Furthermore, a systematic review by Gibbons and colleagues [36] concluded from 116 randomised trials small but significant effects of PROMs as an intervention on HRQoL, but inconclusive impacts on mental health, functioning, and pain.

Moreover, the intervention also showed a positive effect on aftercare health expenditures for patients with hip replacement. The largest factor of the 376.43€ cost savings can be explained by fewer GP visits, fewer prescriptions, and less expenses for remedies such as physiotherapy (see S7 Table). This reduction in GP visits (and consequently receiving less prescriptions and remedies) may potentially be attributed to feeling well taken care of. Patients might have confidence that their recovery is on track when they are not being contacted [37].

Regarding patients with knee replacement, the effects of the intervention were more moderate and only partly significant. The probability of the intervention being dominant was 79.9% when not considering the intervention cost (73.3% with intervention cost). While there were positive outcome effects in terms of PRO-CM and favourable cost effects, they both were nonsignificant considering a 5% significance level (only the comparative Wilcoxon rank-sum test showed significant cost differences between groups). One reason for this difference in relation to patients with hip replacement could be differences in patient characteristics. Patients with knee replacement had a considerably higher BMI and more comorbidities than patients with hip replacement (Table 2), potentially leading to a smaller effect of the intervention, and also on post-surgery health expenditures. For example, a heterogeneity analysis of the PROMoting Quality trial by Langenberger and colleagues [32] showed that the intervention for patients with knee replacement has a larger positive effect for non-obese patients. Moreover, overweight and comorbid patients might visit their GP more often than their healthier counterparts due to their generally worse health status. It can also be assumed that patients with knee replacement need a significantly longer recovery time than patients with hip replacement due to the more complex nature of the surgery. It is therefore reasonable to assume that a longer follow-up period might be appropriate for patients with knee replacement in order to see significant effects. Additionally, patients with knee replacement usually receive closer

postoperative care than patients with hip replacement. This can considerably decrease the impact of the "being taken care of" effect for health expenditures.

This is also confirmed by looking at the break down of the 12-month post-surgery health expenditures (see S8 Table). In patients with knee replacement, we did not observe significant differences in visits for any of the physician categories, and only small differences in cost for outpatient care, remedies, and aids. Nevertheless, our results showed a positive impact of the intervention on overall health expenditures (−375.50€). As for the outcome effect, we saw that the significant effects of the EQ-VAS and the PROMIS-F-SF drive the overall positive but non-significant effect of the PRO-CM in the intervention group. These results correspond with results reported for the whole study population with all patients [31,32].

The study has several strengths. Firstly, the large-scale trial PROMoting Quality provides data from 6,807 patients with hip and knee replacement collected in 9 top tier hospitals in Germany over a period of more than 2 years. Secondly, comprehensive billing data from 24 German statutory health insurances of 1,038 patients was added which allows the longitudinal analysis from the payer's perspective in the first place. Therefore, our CEA draws from a large and carefully collected data set on a high level of detail. The existing data enabled us to include the actual costs incurred per patient into our analyses and to differentiate among different cost categories. Consequently, we investigate cost-effectiveness on actual cost data of included patients, and do not simulate costs as is quite common in other cost-effectiveness analyses [17,18,38]. Lastly, we combined several PROs into a single measure, which considers each instrument, and therefore gives a holistic picture of post-surgery patient-reported health outcome improvement. Additionally, we performed various scenario and sensitivity analyses that increase the interpretative power and robustness of the results.

However, this study also comes with limitations. Firstly, we could only include the 1,038 patients with cost data in the cost-effectiveness analyses. This is less than initially anticipated. With this sample size we could, nevertheless, show significant effects and, hence, rule out potential type 1 errors. As demonstrated for the full sample, in Steinbeck and colleagues [31] and Langenberger and colleagues [32], outcome effects are positive for the intervention group with different effects per health dimension and patient subgroup. Hence, cost-effectiveness might also differ slightly for different samples. As the availability of cost data was not dependent on choice to participate but on the affiliation to a statutory health insurance, we can exclude a potential selection bias. Secondly, the predefined PROM alerts were developed upon a Delphi method and for the assumed average patient with hip or knee replacement. As shown in Kuklinski and colleagues [7], PROM alert thresholds should be defined for individual patient characteristics, in particular their preoperative PROM scores. Therefore, the intervention was not completely tailored to patients' recovery pathways. We hypothesise that with a more patient-tailored approach the effect of the intervention could be increased. Thirdly, we cannot disentangle the effects of the 4 different steps of the intervention, but can only interpret the combined effect of the intervention. Further studies should look at the effects of each step individually.

Evidence from this trial has multiple implications for policy and practice. Firstly, we showed that implementing a PROM monitoring and alert system improves health outcomes for patients with hip and knee replacement in most dimensions. It is highly likely (73.3% to 99.9%) that this health outcome improvement reveals a dominant ICER, even when considering intervention cost. Secondly, hospitals and physicians are able to (1) learn more about patients' post-surgery recovery pathways; (2) monitor them for critical developments; and (3) manage the patient's concern better. Thirdly and even more importantly, the observed decrease of healthcare expenditures means less resource consumption. Thus, in times of rising medical personnel shortages and health services demand, our intervention—and similar

PROM-monitoring interventions—could potentially relieve the healthcare system. To realise such benefits on a large scale, the intervention would need to be implemented into standard care pathways. In Germany, one possibility could be, for example, an implementation as digital health application via the country's fast-track process [39].

In conclusion, as PROMs play a central role in the development of patient-centred care models, this study demonstrated that they can contribute to lower cost and higher patient-reported quality of care. Therefore, health system stakeholders should no longer be reluctant to integrate these measures into standard care pathways.

## Supporting information

**S1 Appendix. Ethics approval, data protection plan, and statistical analysis plan.**
(PDF)

**S2 Appendix. Software description Heartbeat Medical.**
(PDF)

**S1 CONSORT Checklist. CONSORT checklist.**
(DOCX)

**S1 Fig. PROMoting Quality study design.**
(TIF)

**S2 Fig. Mixed effect sensitivity regression analysis—hip patients.**
(TIF)

**S3 Fig. Mixed effect sensitivity regression analysis—knee patients.**
(TIF)

**S4 Fig. Cost-effectiveness plane EQ-5D-5L CEA—hip patients.**
(TIF)

**S5 Fig. Cost-effectiveness plane HOOS-PS CEA—hip patients.**
(TIF)

**S6 Fig. Cost-effectiveness plane EQ-5D-5L CEA—knee patients.**
(TIF)

**S7 Fig. Cost-effectiveness plane KOOS-PS CEA—knee patients.**
(TIF)

**S1 Table. Procedure codes (OPS) for inclusion of patients.**
(DOCX)

**S2 Table. Absolute and relative intervention alert thresholds for each PROM-score.**
(DOCX)

**S3 Table. Critical values and alert reactions.**
(DOCX)

**S4 Table. Properties of the PRO measures.**
(DOCX)

**S5 Table. Calculation and pricing of personnel minutes.**
(DOCX)

**S6 Table. Calculation of the required staff minutes for the intervention.**
(DOCX)

**S7 Table. Post-surgery costs for hip replacement patients.**
(DOCX)

**S8 Table. Post-surgery costs for knee replacement patients.**
(DOCX)

**S9 Table. Descriptive statistics of the whole study sample.**
(DOCX)

## Acknowledgments

Special thanks to all participating consortium members, partner institutions and hospitals, study nurses, and patients.

## Author Contributions

**Conceptualization:** Lukas Schöner, David Kuklinski, Justus Vogel, Christoph Pross, Reinhard Busse, Alexander Geissler.

**Data curation:** Lukas Schöner, David Kuklinski, Laura Wittich, Viktoria Steinbeck, Benedikt Langenberger, Thorben Breitkreuz, Mathias Kretzler, Ursula Marschall.

**Formal analysis:** Lukas Schöner, David Kuklinski.

**Funding acquisition:** Laura Wittich, Justus Vogel, Christoph Pross, Reinhard Busse, Alexander Geissler.

**Investigation:** Lukas Schöner, David Kuklinski.

**Methodology:** Lukas Schöner, David Kuklinski, Thorben Breitkreuz, Justus Vogel, Reinhard Busse, Alexander Geissler.

**Project administration:** Lukas Schöner, David Kuklinski, Laura Wittich, Viktoria Steinbeck, Benedikt Langenberger, Christoph Pross, Reinhard Busse, Alexander Geissler.

**Resources:** Wolfgang Klauser, Mustafa Citak, Georg Matziolis, Daniel Schrednitzki, Kim Grasböck.

**Software:** Felix Compes.

**Supervision:** Reinhard Busse, Alexander Geissler.

**Validation:** Viktoria Steinbeck, Benedikt Langenberger, Thorben Breitkreuz.

**Visualization:** Lukas Schöner.

**Writing – original draft:** Lukas Schöner, David Kuklinski.

**Writing – review & editing:** Lukas Schöner, David Kuklinski, Laura Wittich, Viktoria Steinbeck, Benedikt Langenberger, Thorben Breitkreuz, Felix Compes, Mathias Kretzler, Ursula Marschall, Wolfgang Klauser, Mustafa Citak, Georg Matziolis, Daniel Schrednitzki, Kim Grasböck, Justus Vogel, Christoph Pross, Reinhard Busse, Alexander Geissler.

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
