## [Editor Report · Decision Letter 0]

12 Dec 2023

Dear Dr Schöner, 

Thank you for submitting your manuscript entitled "Cost-effectiveness of a PROM-based remote monitoring and alert intervention for early detection of critical recovery after joint replacement: A randomised controlled trial" for consideration by PLOS Medicine.

Your manuscript has now been evaluated by the PLOS Medicine editorial staff and I am writing to let you know that we would like to send your submission out for external peer review. During the evaluation, we noticed that participant enrollment started on October 1, 2019, while the study was registered at the end of November (11/26/2019). Could you please comment on the late registration?

Please re-submit your manuscript within two working days, i.e. by Dec 14 2023 11:59PM.

Feel free to email me (aschaefer@plos.org) or us at plosmedicine@plos.org if you have any queries relating to your submission.

Kind regards,

Alexandra Schaefer, PhD

Associate Editor

PLOS Medicine

---

## [Decision Letter · Decision Letter 1]

25 Mar 2024

Dear Dr. Schöner,

Thank you very much for submitting your manuscript "Cost-effectiveness of a PROM-based remote monitoring and alert intervention for early detection of critical recovery after joint replacement: A randomised controlled trial" (PMEDICINE-D-23-03623R1) for consideration at PLOS Medicine. 

Your paper was evaluated by an associate editor and discussed among all the editors here. It was also discussed with an academic guest editor with relevant expertise, and sent to independent reviewers, including a statistical reviewer. Thanks again for your patience and understanding during the prolonged review process. The reviews are appended at the bottom of this email and any accompanying reviewer attachments can be seen via the link below:

[LINK]

In light of these reviews, I am afraid that we will not be able to accept the manuscript for publication in the journal in its current form, but we would like to consider a revised version that addresses the reviewers' and editors' comments. Obviously we cannot make any decision about publication until we have seen the revised manuscript and your response, and we plan to seek re-review by one or more of the reviewers. 

Please use the following link to submit the revised manuscript: https://www.editorialmanager.com/pmedicine/

We expect to receive your revised manuscript by Apr 15 2024 . However, if this deadline is not feasible, please contact me by email, and we can discuss a suitable alternative.

Don't hesitate to contact me directly with any questions (aschaefer@plos.org). If you reply directly to this message, please be sure to 'Reply All' so your message comes directly to my inbox.

We look forward to receiving your revised manuscript.

Sincerely,

Alexandra Schaefer, PhD

PLOS Medicine

plosmedicine.org

EDITORIAL COMMENTS

The premise and idea are strong, and a long-term evaluation of arthroplasty using PROMS is a reasonable approach. The background and setting are appropriate and reasonable, the methodology appears correct (in the main), and it is good to see an efficient, partially embedded, registry-style trial. The study is generally rigorous, there is good attention to detail, and the paper is well written and organized.

There are 3 slightly larger concerns:

1) The sample size is good, but there is no a priori power for the expected effect - this is always required for a prospective RCT, even if it is partially embedded.

2) The study is driven from a Health Economics perspective. This has advantages, but also some limitations. The main problem here is that the sample (representative for the study) consists only of those patients with HE data. Since the HE data group is in the minority for the entire registry population, this significantly limits the external validity for any broader interpretation. Therefore, we do not know if the HE data group has different characteristics than the others. This does not make the paper useless by any means (the information and findings are still very valuable - and there is no evidence of differences), but it is a strong caveat to any authoritative conclusion. Please expand or discuss this point to a greater degree and maybe base their conclusions with clear reference to (a population with HE data available only).

3) The restriction and emphasis on HE is also a problem for a wider perspective. It is important to know whether the system is good at picking up events, problems or SAEs. There is no data on this, only the line;

 o [258] “Within 1-year post-surgery adverse events ratios (readmission and reoperation) were generally low and [259] showed no significant differences between control and intervention groups for both procedures.

 o It would therefore be good to see the values and comparative data for this statement in the results section. It is a critical aspect of the proposed system.

***Please note: not all will apply to your paper, but please check each item carefully

GENERAL COMMENTS

1) Please cite the reference numbers in square brackets. Citations should be preceding punctuation.

FINANCIAL DISCLOSURE

The funding statement should include: specific grant numbers, initials of authors who received each award, URLs to sponsors’ websites. Also, please state whether any sponsors or funders (other than the named authors) played any role in study design, data collection and analysis, the decision to publish, or preparation of the manuscript. If they had no role in the research, include this sentence: “The funders had no role in study design, data collection and analysis, decision to publish, or preparation of the manuscript.”

DATA AVAILABILITY

The Data Availability Statement (DAS) requires revision. If the data are not freely available, please describe briefly the ethical, legal, or contractual restriction that prevents you from sharing it. Please also include an appropriate contact (web or email address) for inquiries (this cannot be a study author).

ABSTRACT

1) Please structure your abstract using the PLOS Medicine headings (Background, Methods and Findings, Conclusions).

2) PLOS Medicine requests that main results are quantified with 95% CIs as well as p values. When reporting p values please report as p<0.001 and where higher as the exact p value p=0.002, for example. For the purposes of transparent data reporting, if not including the aforementioned please clearly state the reasons why not. When a p value is given, please specify the statistical test used to determine it.

3) Throughout, suggest reporting statistical information as follows to improve clarity for the reader “22% (95% CI [13%,28%]; p</=)”. Please be sure to define all numerical values at first use. Please amend throughout the abstract and main manuscript. Please note the use of commas to separate upper and lower bounds, as opposed to hyphens as these can be confused with reporting of negative values.

3) Please ensure that all numbers presented in the abstract are present and identical to numbers presented in the main manuscript text.

4) Please include the study design, population and setting, number of participants, years during which the study took place (enrollment and follow up), length of follow up, and main outcome measures.

5) Please specify who was blinded to the intervention and control, define the intervention and control states, provide the number in each group, state that analysis was intention to treat and provide the number of participants lost to follow up in each group.

6) Please include the important dependent variables that are adjusted for in the analyses.

7) Please define all abbreviations including those for statistical reporting at first use.

8) In the last sentence of the Abstract Methods and Findings section, please describe the main limitation(s) of the study's methodology.

9) Please include the clinical trial registry number in the abstract.

AUTHOR SUMMARY

At this stage, we ask that you include a short, non-technical Author Summary of your research to make findings accessible to a wide audience that includes both scientists and non-scientists. The Author Summary should immediately follow the Abstract in your revised manuscript. This text is subject to editorial change and should be distinct from the scientific abstract. Ideally each sub-heading should contain 2-3 single sentence, concise bullet points containing the most salient points from your study. In the final bullet point of ‘What Do These Findings Mean?’, please include the main limitations of the study in non-technical language. Please see our author guidelines for more information: https://journals.plos.org/plosmedicine/s/revising-your-manuscript#loc-author-summary

METHODS AND RESULTS

1) PLOS Medicine requests that main results are quantified with 95% CIs as well as p values. We suggest reporting statistical information as detailed above – see under ABSTRACT

2) Please present numerators and denominators for percentages (at least in the Tables [not necessarily each time they're mentioned]).

3) Please complete the CONSORT checklist and ensure that all components of CONSORT are present in the manuscript, including [how randomization was performed, allocation concealment, blinding of intervention, definition of lost to follow-up, power statement]. When completing the checklist, please use section and paragraph numbers, rather than page numbers.

4) Please include the study protocol document and analysis plan, with any amendments, as Supporting Information to be published with the manuscript if accepted.

5) Please present the safety data for the study including numbers of specific events and whether or not adverse events are thought to be related to the intervention.

DISCUSSION

Please present and organize the Discussion as follows: a short, clear summary of the article's findings; what the study adds to existing research and where and why the results may differ from previous research; strengths and limitations of the study; implications and next steps for research, clinical practice, and/or public policy; one-paragraph conclusion (no subheading).

FIGURES AND TABLES 

1) Please provide titles and legends for all figures and tables (including those in Supporting Information files). 

2) Please define all abbreviations used in each figure/table (including those in Supporting Information files). 

3) Please consider avoiding the use of red and green in order to make your figure more accessible to those with color blindness. 

SUPPLEMENTARY MATERIAL

1) We suggest reporting statistical information as detailed above – see under ABSTRACT. Please be sure to define all numerical values.

2) As for the main manuscript, please indicate whether analyses are adjusted to help facilitate transparent data reporting please also detail the factors adjusted for and present the unadjusted analyses for comparison. If not, please clearly state the reasons why not.

3) Please cite your Supporting Information as outlined here: https://journals.plos.org/plosmedicine/s/supporting-information

REFERENCES

1) PLOS uses the numbered citation (citation-sequence) method and first six authors, et al.

2) Please ensure that journal name abbreviations match those found in the National Center for Biotechnology Information (NCBI) databases (http://www.ncbi.nlm.nih.gov/nlmcatalog/journals), and are appropriately formatted and capitalised.

3) Where website addresses are cited, please specify the date of access (e.g. [accessed: 16/09/2023]).

4) Please also see https://journals.plos.org/plosmedicine/s/submission-guidelines#loc-references for further details on reference formatting. 

Comments from the reviewers:

Reviewer #1: This RCT study assessed the cost-effectiveness of a patient-reported outcome measures (PROMs)-based monitoring intervention for hip and knee replacement patients. The intervention aimed for early detection of critical recovery paths. Results indicated improved outcomes and lower costs for the intervention group, with a dominant ICER for both hip and knee replacements. The results suggest that the intervention is cost-effective instrument for remote monitoring of patients undergoing HR or KR. Overall, the study is well-designed and conducted, and the results hold significant implications for clinical practice and health policy. Below are my specific comments. 

1. Data cannot be publicly disclosed due to restrictions, and the authors CP and FC's affiliations with private medical device and solution companies warrant further examination.

2. Title: It might be better to avoid using acronyms (PROM) in the title. 

3. Abstract: It would be better to provide the full name of PROM when it first appears in the abstract. 

4. Line 68-70, introduction: The numbers are given, but there is little discussion of their implications or how they compare to other regions or time periods. It would be helpful to provide a brief comparison of these statistics to give the reader a sense of how they compare to other countries or regions.

5. Line 80-81, introduction: the authors have identified the gap in the literature but could further explain how this gap impacts clinical practice and policy decisions to underscore the importance of the research.

6. Line 95-96, Method: What do ASA categories 4-6 signify? Are those severe cases? Providing detailed explanations of these categories would help readers better understand the rationale for the exclusion criteria. 

7. Line 96-97, Method: I am curious what is the proportion of patients who lack an email account or the ability to use digital PROMs? Given that patients undergoing HR or KR are mostly elderly individuals, who may not be adept with modern technology, there are concerns regarding the representativeness of the patient sample included in this study.

8. Line 106-107, Method: It would be beneficial to include a description of Heartbeat ONE software. Clarifying what this software is and its functions would help readers understand how it is used within the context of the study.

9. Line 113-126, Method: it is mentioned that patients in the intervention group received additional follow-ups at 1, 3, and 6 months. There is a concern that these extra check-ins could influence the PROMs due to a potential perception among patients of receiving increased attention and care.

10. Line 144-146: Method: It would be better for the authors to provide a more detailed explanations regarding the methodology used to standardize various scales into a unified scoring system ranging from 1 to 100.

11. Line 182, Method: It would be better to include a succinct explanation of the 'within-trial cost-effectiveness approach' used in their study, or at least, cite a reference for readers seeking a more comprehensive understanding of this methodology.

12. Line 188-190, Method: A detailed explanation would be beneficial regarding the impact of a 95% winsorisation on the cost data and the rationale behind choosing this specific threshold. Considering that costs are the primary outcome of interest, replacing extreme values with thresholds could notably influence the results. An elaboration on why a 95% winsorisation was preferred over, say, a 99% threshold, would enhance the reader's understanding of the methodological choices.

Reviewer #2: Overall, this is a well written manuscript on a very important area. The study design and methods that are used are robust. There is a minor concern regarding the data analytical methods that were used, particularly mixed-models. While the investigators used clustering effect of hospitals in modeling, it is not clear how the temporal effect on outcomes/cost was taken into account. In the revised version, the authors should provide revised analytical approach taking in account the time effect.

Reviewer #3: This is a report of the effectiveness and cost-effectiveness of a PROM intervention to detect and respond to issues following hip or knee surgery.

The manuscript would benefit from attention to the following points:

 1. A novel trial idea, and patient-centred outcome for which the authors should be commended.

 2. Although the authors stated they followed CONSORT for reporting, I could not find a copy of the CONSORT checklist attached with page numbers referencing each point in the manuscript. This would be extremely helpful in ensuring all relevant checklist points about trial conduct were addressed and described.

 3. The choice of trial outcome is unusual and somewhat hard to follow. First a composite PROM index was constructed (without validation), and this was combined with costs for an incremental cost-effectiveness ratio. An effect size of (0.15 standard deviation) from the mean in one of the PROMs, or the composite was used, according to the Appendix in the protocol paper. There was no effect size given for costs, or for cost-effectiveness.

4. Could the authors justify why they chose not to have a primary end point of the trial as a minimally important difference in the PRO, and a secondary outcome of cost-effectiveness? E.g. with an ICER of cost per additional person achieving a meaningful PRO improvement. The primary outcome is not well articulated, and a willingness to pay threshold for the German payer was not provided.

5. Trial and cost-effectiveness sample size estimation. Could the authors please outline the effect size for PRO-CM, effect size for costs, and overall effect size for cost-effectiveness they were testing? It is unclear from Appendix II.

6. Please provide the trial registration number on e.g. Clinicaltrials.govdatabase or similar

7. Why was the included sample in this analysis so low, one sixth of the whole randomised population? 

8. If participation was limited to single health insurer. How were these patients different or similar to the whole trial population in terms of SES, age, sex, region, income, occupation etc. Very large potential for selection bias.

9. Were the PRO results adjusted for baseline values in the between group comparisons? 

10. The results indicate the intervention worked in hips but not knees. Could the authors re-phrase their interpretation in line with the results, and comment on why this may be the case.

11. In economic jargon, a dominant or dominated intervention is not usually reported as an ICER, because negative ratios can be difficult to interpret.

[LINK]

General journal requests:

---

## [Decision Letter · Decision Letter 2]

24 Jun 2024

Dear Dr Schöner,

Many thanks for submitting your manuscript "Cost-effectiveness of a patient-reported outcome based remote monitoring and alert intervention for early detection of critical recovery after joint replacement: A randomised controlled trial" (PMEDICINE-D-23-03623R2) to PLOS Medicine. The paper has been seen again by a statistician and we have secured an additional review with a focus on health economics; their comments are included below and can also be accessed here: [LINK]

As you will see, there a still number of questions about specific study details, including the primary study outcome and the calculation of the "probability of cost-effectiveness". After discussing the paper with the editorial team and an academic editor with relevant expertise, we ask you to carefully address the comments in a further revision. We plan to send the revised paper to some or all of the original reviewers.

We ask that you submit your revision by Jul 15 2024. However, if this deadline is not feasible, please contact me by email, and we can discuss a suitable alternative.

Don't hesitate to contact me directly with any questions (atosun@plos.org). 

Best regards, 

Alex 

Alexandra Tosun, PhD 

Associate Editor

PLOS Medicine

atosun@plos.org

Comments from the reviewers: 

Reviewer #1: The authors have adequately answered all my questions and questions from other reviewers, and have revised the paper accordingly. In my opinion, this paper is acceptable in its current form. 

Reviewer #4: Thank you to the authors for submitting this paper on an interesting topic. I have concentrated on the statistical and health economics aspect of the study, hopefully some of the other reviewers can comment on the intervention and clinical context of the study. I have several major comments, most of which seem to have been picked up by the previous reviewers.

PRO-CM

I've simply not convinced by the use of the PRO-CM as the primary outcome measure. Combining so many different measures together and then standardising them creates something that in my opinion is essentially meaningless. What does a unit change in this PRO-CM actually mean? I know that 0.15 SD change seems to have been quoted in the protocol, but this seems to be completely arbitrary. I would strongly suggest using another measure (either a generic preference based measure such as the EQ-5D-5L or a condition-specific measure) as the primary outcome and having this composite measure as an experimental secondary outcome. To me there is simply not enough precedent to use this in the ICER and present it as the primary economic outcome. 

Probability of Cost-Effectiveness

Calculating the "probability of cost effectiveness" from an ICER within a Bayesian cost effectiveness framework explicitly requires a threshold to compare the ICER to. Without a cost effectiveness threshold for the ICER for the PRO-CM, the authors use the proportion of bootstrapped estimates that have negative incremental costs. This is incorrect and should be dropped from the paper as it is misleading. It's essentially implying the threshold is 0. 

Dominant and Dominated Interventions

As picked up by the reviewers, ICERs for dominant and dominated interventions don't have any meaning and should be removed from all parts of the paper, not just the abstract.

Controlling for Baseline PRO-CM

As picked up by one of the other reviewers, adjusting for baseline PRO-CM could be a good idea. Given the nature of the results and the intervention, I would like to see a sensitivity analysis with baseline PRO-CM included in the mixed effect regression models. This has been good practice in within-trial economic evaluations for around 20 years, please see Manca, Hawkins & Sculphur (2004): https://doi.org/10.1002/hec.944

QALYs/CUA

As the authors collected EQ-5D-5L, I'm confused why they didn't consider presenting a CUA alongside the CEA?

* Please upload any figures associated with your paper as individual TIF or EPS files with 300dpi resolution at resubmission; please read our figure guidelines for more information on our requirements: http://journals.plos.org/plosmedicine/s/figures. While revising your submission, please upload your figure files to the PACE digital diagnostic tool, https://pacev2.apexcovantage.com/. PACE helps ensure that figures meet PLOS requirements. To use PACE, you must first register as a user. Then, login and navigate to the UPLOAD tab, where you will find detailed instructions on how to use the tool. If you encounter any issues or have any questions when using PACE, please email us at PLOSMedicine@plos.org.

---

## [Decision Letter · Decision Letter 3]

5 Aug 2024

Dear Dr. Schöner,

Thank you very much for re-submitting your manuscript "Cost-effectiveness of a patient-reported outcome based remote monitoring and alert intervention for early detection of critical recovery after joint replacement: A randomised controlled trial" (PMEDICINE-D-23-03623R3) for review by PLOS Medicine.

Thank you for your detailed response to the editors' and reviewers' comments. I have discussed the paper with my colleagues and the academic editor, and it has also been seen again by one of the original reviewers. The changes made to the paper were mostly satisfactory to the reviewer. As such, we intend to accept the paper for publication, pending your attention to the reviewer and editorial comments below in a further revision. When submitting your revised paper, please once again include a detailed point-by-point response to the editorial comments.

[LINK]

In revising the manuscript for further consideration here, please ensure you address the specific points made by each reviewer and the editors. In your rebuttal letter you should indicate your response to the reviewers' and editors' comments and the changes you have made in the manuscript. Please submit a clean version of the paper as the main article file. A version with changes marked must also be uploaded as a marked up manuscript file. Please also check the guidelines for revised papers at http://journals.plos.org/plosmedicine/s/revising-your-manuscript for any that apply to your paper. 

We ask that you submit your revision within 1 week (Aug 12 2024). However, if this deadline is not feasible, please contact me by email, and we can discuss a suitable alternative.

Please do not hesitate to contact me directly with any questions (atosun@plos.org). If you reply directly to this message, please be sure to 'Reply All' so your message comes directly to my inbox.

We look forward to receiving the revised manuscript.

Sincerely,

Alexandra Tosun, PhD

Associate Editor 

PLOS Medicine

plosmedicine.org

Requests from Editors:

GENERAL

We feel that the manuscript could be improved with respect to structure and clarity. When revising, please keep in mind that the manuscript should be accessible to readers who may not be familiar with the topic.

We suggest removing the abbreviations "KR" and "HR" and spelling them out throughout the manuscript. 

PLOS Medicine prefers the use of patient-centered language, e.g. patients undergoing hip replacement and patients undergoing knee replacement (or similar). Please revise throughout the manuscript, including tables and figures (including those in the Supporting Information).

We have noticed that you refer to results that are not statistically significant as insignificant. Please note that insignificant usually implies unimportance, without statistical connotations. Please revise and change to 'non-significant' throughout.

DATA AVAILABILITY

The Data Availability Statement (DAS) requires revision. Please include an appropriate contact (web or email address) for inquiries (this cannot be a study author).

ABSTRACT

1) Please combine the Methods and Findings sections into one section.

2) l.39: “3697 hip and 3110 knee replacement patients” - We prefer the use of patient-centered language (i.e. “3697 patients with hip replacement and 3110 patients with knee replacement”). Please revise accordingly throughout the Abstract and the main text.

3) l.41: Please add details about the randomization, e.g. 1:1.

4) l.46ff: Per CONSORT, please note that only the primary outcome of the trial should be reported in your Abstract. Secondary outcomes should only be included in the Abstract if all secondary outcomes are fully reported. For trials that have many secondary outcomes, the Abstract should be limited to reporting the primary outcome.

5) l.47ff: Please define abbreviations at first use, such as HOOS-PS, KOOS-PS or CI.

6) In the last sentence of the Abstract Methods and Findings section, please describe the main limitation(s) of the study's methodology.

7) ll.56-57: The term "tendency" is used to refer to a nonsignificant P value, please remove and revise the conclusions accordingly. 

8) Please remove the Funding Statement from the Abstract.

AUTHOR SUMMARY

1) Please see the comments regarding patient-centered language and the term ‘tendency’ under ABSTRACT and revise accordingly.

2) In the final bullet point of ‘What Do These Findings Mean?’ Please include the main limitations of the study in non-technical language.

INTRODUCTION

1) l.105: Please define OECD.

2) ll.115-116, please change to: In further oncological studies, it was shown that the intervention was cost-effective [17,18].

METHODS AND RESULTS

1) ll.138-149: Please outline the main inclusion/exclusion criteria in the main text. You may then refer to the detailed criteria in the Supplementary Material and the published protocol.

2) l.162: “Patients in the control group received the standard of care.” – We suggest a brief outline of what SOC entails, as a global audience may have different experiences with SOC after hip and knee replacement surgery.

3) ll.160-172: We suggest revising this paragraph for clarity and structure.

4) l.193: Please define ‘SD’ at first use.

5) l.214: Please provide details on the power calculation and the statistical analysis plan in the main text.

6) l.224: The terms gender and sex are not interchangeable (as discussed in https://www.who.int/health-topics/gender); please use the appropriate term.

7) l.225: Please define ‘BMI’ at first use.

8) l.232: Please define ‘CI’ at first use.

9) l.259, please change to: “We focused… that only differed…”

10) Table 1: Please define ‘PROM’.

11) l.265ff: Please revise for tense. Also, please temper claims of primacy of results by stating, "to our knowledge" or something similar (i.e. Since the application of the PRO-CM as primary outcome parameter was, to our knowledge, novel and…).

12) l.267: Please define ‘CEA’ at first use.

13) We feel that it may be confusing to readers as to what the primary outcome of the study is. In lines 186-187 you state that you are using a PRO composite measure (PRO-CM) as the primary outcome, which does not mention the cost-effectiveness component, while in line 249 you specify the study's main outcome, i.e., the incremental cost-effectiveness ratio (ICER). We suggest revising the manuscript to improve clarity and suggest that under "Outcomes" you first clearly state the primary outcome, and the secondary outcomes followed by a detailed description of these (as already done). 

14) Figure 1: Please define ‘PROM’. Please revise with regard to the use of patient-centered language. Also, please change “patients excluded without consent or with non-participating health insurance” to “patients excluded due to lack of consent or non-participating health insurance”.

15) l.285: Please remove the word ‘respectively’.

16) l.286: In brackets, please add how ‘obese’ was defined.

17) ll.285-286, please change to: 32.8% of patients undergoing hip replacement and 53.1% of patients undergoing knee replacement were obese.

18) ll.286-287, please change to: 38.4% of the patients who underwent hip replacement surgery, but only 29.0% of the patients who underwent knee replacement surgery had no comorbidities recorded.

19) Table 2: Please exchange ‘gender’ with ‘sex’.

20) l.291ff: We suggest describing the results throughout as follows: “Comparative analyses showed that, for patients undergoing hip replacement, the PRO-CM [26] was significantly higher…”

21) l.293: Please include statistical information when reporting numerical values (i.e. “51.22 SD (9.66)”).

22) ll.293-294: Please rephrase (The intervention group did not improve, but the outcome measure was significantly higher in the intervention group).

23) Table 3: Please spell out ‘y’ and ‘Adj’. We also suggest spelling out ‘IG’ and CG’ and adding a definition for ‘Incremental’ below the table.

24) l.296ff: Please revise for tense (These findings were supported…).

25) Figure 2: Please include the relevant abbreviations in the figure description and provide exact p-values instead of asterisks with significance levels. It currently appears as if "Hip/Knee replacement patients" are the y-axis labels. Assuming these serve as headings, please move them above the graph and adjust them using patient-centered language.

26) l.333: The term "tendency" is used to refer to a nonsignificant P value, please remove and revise accordingly.

27) ll.342-343: “The mixed effect model, however, only shows a weakly significant intervention effect on the KR costs.” – We suggest adding a reference to Figure 2.

28) Figure 3: Please provide a unit for cost. Please define ‘PROM’.

29) l.371ff: Please revise for tense (e.g. “The findings from our base case models were supported by the sensitivity analyses.”)

30) l.380: The term "trend" is used to refer to a nonsignificant P value. The term trend should be used only when the test for trend has been conducted. Please revise accordingly.

DISCUSSION

1) l.408: Please rephrase “Main effects of the intervention for the total sample” to avoid referring to participants as samples. Please revise throughout the main text.

2) l.412ff: The terms gender and sex are not interchangeable (as discussed in https://www.who.int/health-topics/gender); please use the appropriate term and revise throughout.

3) ll.430-433: We feel that the statements are repetitive of each other, please revise.

4) l.452ff: When discussing results, please use past tense. 

5) l.498, please change to: “study demonstrated”

REFERENCES

Please ensure that journal name abbreviations match those found in the National Center for Biotechnology Information (NCBI) databases (http://www.ncbi.nlm.nih.gov/nlmcatalog/journals), and are appropriately formatted and capitalised. For example, “Patient Related Outcome Measures” in reference [2] should be “Patient Relat Outcome Meas”.

SUPPLEMENTARY MATERIAL

1) Thank you for providing the completed CONSORT checklist. Please replace the page numbers with paragraph numbers per section (e.g. "Methods, paragraph 1"), since the page numbers of the final published paper may be different from the page numbers in the current manuscript.

2) In the published article, supporting information files are accessed only through a hyperlink attached to the captions. For this reason, you must list captions at the end of your manuscript file. You may include a caption within the supporting information file itself, as long as that caption is also provided in the manuscript file. Do not submit a separate caption file.

When SI files are contained with a single file:

Please label the file as ‘S1 Supporting Information’.

Please apply alphabetical labelling to each table and figure contained within the S1 file. For example, ‘Fig A’ to ‘Fig Z’ and ‘Table A’ to ‘Table Z’.

Plain text does not need to be labelled and can just be given a title as necessary. For example, ‘Statistical Analysis Plan’.

Please cite tables/figures as ‘Fig A in S1 Supporting Information’ and/or ‘Table A in S1 Supporting Information’, for example.

Please cite plain text as, ‘Statistical Analysis Plan in S1 Supporting Information’, for example.

When SI files are uploaded as separate files:

Please label tables as ‘S1 Table’ (so on) and figures as ‘S1 Fig’ (and so on).

Any additional documents (protocols/analysis plans etc.) can be labelled as ‘S1 Protocol’, for example. Please cite items as exactly as labelled.

3) Please revise the Supplementary Material according to the comments above. Please note that the Supplementary Material will be published as provided by the authors.

SOCIAL MEDIA

To help us extend the reach of your research, please provide any X (formerly known as Twitter) handle(s) that would be appropriate to tag, including your own, your co-authors’, your institution, funder, or lab. Please enter in the submission form any handles you wish to be included when we post about this paper.

Comments from Reviewers:

Reviewer #4: Thank you to the authors for responding so comprehensively to my comments and making the requisite changes to manuscript.

I think the compromise of retaining the primary outcome measure (to be in line with the protocol) but adding the secondary outcomes is fair. 

I think the justification for the 0.15SD is fair, however I would like an extra sentence referencing Teare et al (2014) explaining this in the manuscript before publication. 

I'm still not convinced how useful the "probability of dominance" is as a metric, but don't think it is a major issue as it is just an output from Figure 3. If the authors think they should retain it, then that is fine with me. 

Thank you for removing the text regarding negative ICERs, and including the extra analysis which includes controlling for baseline EQ-5D-5L. 

The justification regarding not conducting a CUA is of course correct (my mistake!)

If the authors add the extra sentence regarding the justification of the 0.15SD effect size, I am happy to approve the paper for publication, and don't need to review it again. Good luck to the authors with this one!

[LINK]

General Editorial Requests

---

## [Editor Report · Decision Letter 4]

14 Aug 2024

Dear Dr Schöner, 

On behalf of my colleagues and the Guest Academic Editor, David Beard, I am pleased to inform you that we have agreed to publish your manuscript "Cost-effectiveness of a patient-reported outcome based remote monitoring and alert intervention for early detection of critical recovery after joint replacement: A randomised controlled trial" (PMEDICINE-D-23-03623R4) in PLOS Medicine.

I appreciate your thorough responses to reviewers' and editors' comments, and your patience throughout the editorial process. We look forward to publishing your manuscript, and editorially there are only a few remaining minor stylistic points that should be addressed prior to publication. We will carefully check whether these changes have been made. If you have any questions or concerns regarding these final requests, please feel free to contact me at atosun@plos.org.

Please see below the minor points that we request you respond to:

1) Abstract: "For effect evaluation, a PROM-based composite measure (PRO-CM) was developed that combines changes in different PROMs into an index ranging from 0 to 100". - We think it would be helpful to mention the exact number of PROMs included and briefly describe the types of PROMs included in the PRO-CM (adding the full list of PROMs would be too extensive). For example: The PRO-CM included 6 PROMs focused on quality of life and various aspects of physical and mental health.

2) Abstract: ll.55-56, please change to: “Further it showed a non-significant cost reduction in knee replacement patients.”

3) Results: l.362: As above, please remove the wording “weakly significant” (the results are either statistically significant or not).

PRESS

Sincerely, 

Alexandra Tosun, PhD 

Associate Editor 

PLOS Medicine